# Alias-Free Generative Adversarial Networks

**Tero Karras**
NVIDIA
tkarras@nvidia.com

**Miika Aittala**
NVIDIA
maittala@nvidia.com

**Samuli Laine**
NVIDIA
slaine@nvidia.com

**Erik Härkönen**[*]
Aalto University and NVIDIA
erik.harkonen@aalto.fi

**Janne Hellsten**
NVIDIA
jhellsten@nvidia.com

**Jaakko Lehtinen**
NVIDIA and Aalto University
jlehtinen@nvidia.com

**Timo Aila**
NVIDIA
taila@nvidia.com

## Abstract

We observe that despite their hierarchical convolutional nature, the synthesis process of typical generative adversarial networks depends on absolute pixel coordinates in an unhealthy manner. This manifests itself as, e.g., detail appearing to be glued to image coordinates instead of the surfaces of depicted objects. We trace the root cause to careless signal processing that causes aliasing in the generator network. Interpreting all signals in the network as continuous, we derive generally applicable, small architectural changes that guarantee that unwanted information cannot leak into the hierarchical synthesis process. The resulting networks match the FID of StyleGAN2 but differ dramatically in their internal representations, and they are fully equivariant to translation and rotation even at subpixel scales. Our results pave the way for generative models better suited for video and animation.

## 1 Introduction

The resolution and quality of images produced by generative adversarial networks (GAN) [19] have seen rapid improvement recently [27, 11, 29, 30]. They have been used for a variety of applications, including image editing [42, 47, 37, 20, 34, 3], domain translation [62, 32, 53, 36], and video generation [49, 15, 21]. While several ways of controlling the generative process have been found [8, 26, 10, 36, 22, 2, 7, 41, 6], the foundations of the synthesis process remain only partially understood.

In the real world, details of different scale tend to transform hierarchically. For instance, moving a head causes the nose to move, which in turn moves the skin pores on it. The structure of a typical GAN generator is analogous: coarse, low-resolution features are hierarchically refined by upsampling layers, locally mixed by convolutions, and new detail is introduced through nonlinearities. We observe that despite this superficial similarity, current GAN architectures do not synthesize images in a natural hierarchical manner: the coarse features mainly control the *presence* of finer features, but not their precise positions. Instead, much of the fine detail appears to be fixed in pixel coordinates. This disturbing "texture sticking" is clearly visible in latent interpolations (see Figure 1 and our accompanying videos on the project page https://nvlabs.github.io/stylegan3), breaking the illusion of a solid and coherent object moving in space. Our goal is an architecture that

---

[*]This work was done during an internship at NVIDIA.

35th Conference on Neural Information Processing Systems (NeurIPS 2021).

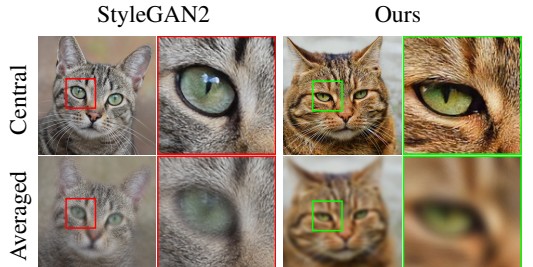
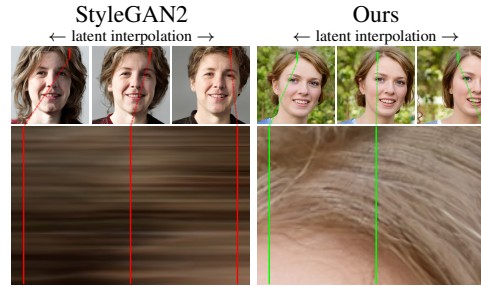

Figure 1: Examples of "texture sticking". **Left:** The average of images generated from a small neighborhood around a central latent (top row). The intended result is uniformly blurry because all details should move together. However, with StyleGAN2 many details (e.g., fur) stick to the same pixel coordinates, showing unwanted sharpness. **Right:** From a latent space interpolation (top row), we extract a short vertical segment of pixels from each generated image and stack them horizontally (bottom). The desired result is hairs moving in animation, creating a time-varying field. With StyleGAN2 the hairs mostly stick to the same coordinates, creating horizontal streaks instead.

exhibits a more natural transformation hierarchy, where the exact sub-pixel position of each feature is exclusively inherited from the underlying coarse features.

It turns out that current networks can partially bypass the ideal hierarchical construction by drawing on unintentional positional references available to the intermediate layers through image borders [25, 31, 58], per-pixel noise inputs [29] and positional encodings, and aliasing [5, 61]. Aliasing, despite being a subtle and critical issue [38], has received little attention in the GAN literature. We identify two sources for it: 1) faint after-images of the pixel grid resulting from non-ideal upsampling filters[2] such as nearest, bilinear, or strided convolutions, and 2) the pointwise application of nonlinearities such as ReLU [52] or swish [40]. We find that the network has the means and motivation to amplify even the slightest amount of aliasing and combining it over multiple scales allows it to build a basis for texture motifs that are fixed in screen coordinates. This holds for all filters commonly used in deep learning [61, 51], and even high-quality filters used in image processing.

How, then, do we eliminate the unwanted side information and thereby stop the network from using it? While borders can be solved by simply operating on slightly larger images, aliasing is much harder. We begin by noting that aliasing is most naturally treated in the classical Shannon-Nyquist signal processing framework, and switch focus to bandlimited functions on a continuous domain that are merely represented by discrete sample grids. Now, successful elimination of all sources of positional references means that details can be generated equally well regardless of pixel coordinates, which in turn is equivalent to enforcing continuous *equivariance* to sub-pixel translation (and optionally rotation) in all layers. To achieve this, we describe a comprehensive overhaul of all signal processing aspects of the StyleGAN2 generator [30]. Our contributions include the surprising finding that current upsampling filters are simply not aggressive enough in suppressing aliasing, and that extremely high-quality filters with over 100dB attenuation are required. Further, we present a principled solution to aliasing caused by pointwise nonlinearities [5] by considering their effect in the continuous domain and appropriately low-pass filtering the results. We also show that after the overhaul, a model based on 1×1 convolutions yields a strong, rotation equivariant generator.

Once aliasing is adequately suppressed to force the model to implement more natural hierarchical refinement, its mode of operation changes drastically: the emergent internal representations now include coordinate systems that allow details to be correctly attached to the underlying surfaces. This promises significant improvements to models that generate video and animation. The new StyleGAN3 generator matches StyleGAN2 in terms of FID [23], while being slightly heavier computationally. Our implementation and pre-trained models are available at `https://github.com/NVlabs/stylegan3`

Several recent works have studied the lack of translation equivariance in CNNs, mainly in the context of classification [25, 31, 58, 5, 33, 61, 12, 63, 51]. We significantly expand upon the antialiasing

---

[2]Consider nearest neighbor upsampling. If we upsample a 4×4 image to 8×8, the original pixels will be clearly visible, allowing one to reliably distinguish between even and odd pixels. Since the same is true on all scales, this (leaked) information makes it possible to reconstruct even the absolute pixel coordinates. With better filters such as bilinear or bicubic, the clues get less pronounced, but are nevertheless evident for the generator.

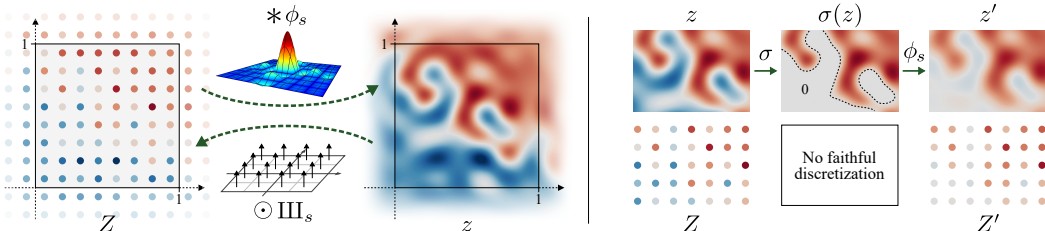

Figure 2: **Left:** Discrete representation $Z$ and continuous representation $z$ are related to each other via convolution with ideal interpolation filter $\phi_s$ and pointwise multiplication with Dirac comb $III_s$. **Right:** Nonlinearity $\sigma$, ReLU in this example, may produce arbitrarily high frequencies in the continuous-domain $\sigma(z)$. Low-pass filtering via $\phi_s$ is necessary to ensure that $Z'$ captures the result.

measures in this literature and show that doing so induces a fundamentally altered image generation behavior. Group-equivariant CNNs aim to generalize the efficiency benefits of translational weight sharing to, e.g., rotation [16, 57, 55, 54] and scale [56]. Our $1 \times 1$ convolutions can be seen an instance of a continuously E(2)-equivariant model [54] that remains compatible with, e.g., channel-wise ReLU nonlinearities and modulation. Dey et al. [17] apply $90°$ rotation-and-flip equivariant CNNs [16] to GANs and show improved data efficiency. Our work is complementary, and not motivated by efficiency. Recent implicit network [45, 48, 13] based GANs [4, 46] generate each pixel independently via similar $1 \times 1$ convolutions. While equivariant, these models do not help with texture sticking, as they do not use an upsampling hierarchy or implement a shallow non-antialiased one.

## 2 Equivariance via continuous signal interpretation

To begin our analysis of equivariance in CNNs, we shall first rethink our view of what exactly is the signal that flows through a network. Even though data may be stored as values in a pixel grid, we cannot naïvely hold these values to directly represent the signal. Doing so would prevent us from considering operations as trivial as translating the contents of a feature map by half a pixel.

According to the Nyquist–Shannon sampling theorem [44], a regularly sampled signal can represent any continuous signal containing frequencies between zero and half of the sampling rate. Let us consider a two-dimensional, discretely sampled feature map $Z[\boldsymbol{x}]$ that consists of a regular grid of Dirac impulses of varying magnitudes, spaced $1/s$ units apart where $s$ is the sampling rate. This is analogous to an infinite two-dimensional grid of values.

Given $Z[\boldsymbol{x}]$ and $s$, the Whittaker–Shannon interpolation formula [44] states that the corresponding continuous representation $z(\boldsymbol{x})$ is obtained by convolving the discretely sampled Dirac grid $Z[\boldsymbol{x}]$ with an ideal interpolation filter $\phi_s$, i.e., $z(\boldsymbol{x}) = \big(\phi_s * Z\big)(\boldsymbol{x})$, where $*$ denotes continuous convolution and $\phi_s(\boldsymbol{x}) = \text{sinc}(sx_0) \cdot \text{sinc}(sx_1)$ using the signal processing convention of defining $\text{sinc}(x) = \sin(\pi x)/(\pi x)$. $\phi_s$ has a bandlimit of $s/2$ along the horizontal and vertical dimensions, ensuring that the resulting continuous signal captures all frequencies that can be represented with sampling rate $s$.

Conversion from the continuous to the discrete domain corresponds to sampling the continuous signal $z(\boldsymbol{x})$ at the sampling points of $Z[\boldsymbol{x}]$ that we define to be offset by half the sample spacing to lie at the "pixel centers", see Figure 2, left. This can be expressed as a pointwise multiplication with a two-dimensional Dirac comb $III_s(\boldsymbol{x}) = \sum_{X \in \mathbb{Z}^2} \delta\big(\boldsymbol{x} - (X + \frac{1}{2})/s\big)$.

We earmark the unit square $\boldsymbol{x} \in [0, 1]^2$ in $z(\boldsymbol{x})$ as our canvas for the signal of interest. In $Z[\boldsymbol{x}]$ there are $s^2$ discrete samples in this region, but the above convolution with $\phi_s$ means that values of $Z[\boldsymbol{x}]$ outside the unit square also influence $z(\boldsymbol{x})$ inside it. Thus storing an $s \times s$ -pixel feature map is not sufficient; in theory, we would need to store the entire infinite $Z[\boldsymbol{x}]$. As a practical solution, we store $Z[\boldsymbol{x}]$ as a two-dimensional array that covers a region slightly larger than the unit square (Section 3.2).

Having established correspondence between bandlimited, continuous feature maps $z(\boldsymbol{x})$ and discretely sampled feature maps $Z[\boldsymbol{x}]$, we can shift our focus away from the usual pixel-centric view of the signal. In the remainder of this paper, we shall interpret $z(\boldsymbol{x})$ as being the actual signal being operated on, and the discretely sampled feature map $Z[\boldsymbol{x}]$ as merely a convenient encoding for it.

**Discrete and continuous representation of network layers**    Practical neural networks operate on the discretely sampled feature maps. Consider operation $\mathbf{F}$ (convolution, nonlinearity, etc.) operating on a discrete feature map: $Z' = \mathbf{F}(Z)$. The feature map has a corresponding continuous counterpart, so we also have a corresponding mapping in the continuous domain: $z' = \mathbf{f}(z)$. Now, an operation specified in one domain can be seen to perform a corresponding operation in the other domain:

$$\mathbf{f}(z) = \phi_{s'} * \mathbf{F}(\text{Ш}_s \odot z), \qquad\qquad \mathbf{F}(Z) = \text{Ш}_{s'} \odot \mathbf{f}(\phi_s * Z), \qquad\qquad (1)$$

where $\odot$ denotes pointwise multiplication and $s$ and $s'$ are the input and output sampling rates. Note that in the latter case $\mathbf{f}$ must not introduce frequency content beyond the output bandlimit $s'/2$.

## 2.1   Equivariant network layers

Operation $\mathbf{f}$ is equivariant with respect to a spatial transformation $\mathbf{t}$ of the 2D plane if it commutes with it in the continuous domain: $\mathbf{t} \circ \mathbf{f} = \mathbf{f} \circ \mathbf{t}$. We note that when inputs are bandlimited to $s/2$, an equivariant operation must not generate frequency content above the output bandlimit of $s'/2$, as otherwise no faithful discrete output representation exists.

We focus on two types of equivariance in this paper: translation and rotation. In the case of rotation the spectral constraint is somewhat stricter — rotating an image corresponds to rotating the spectrum, and in order to guarantee the bandlimit in both horizontal and vertical direction, the spectrum must be limited to a disc with radius $s/2$. This applies to both the initial network input as well as the bandlimiting filters used for downsampling, as will be described later.

We now consider the primitive operations in a typical generator network: convolution, upsampling, downsampling, and nonlinearity. Without loss of generality, we discuss the operations acting on a single feature map: pointwise linear combination of features has no effect on the analysis.

**Convolution**    Consider a standard convolution with a discrete kernel $K$. We can interpret $K$ as living in the same grid as the input feature map, with sampling rate $s$. The discrete-domain operation is simply $\mathbf{F}_{\text{conv}}(Z) = K * Z$, and we obtain the corresponding continuous operation from Eq. 1:

$$\mathbf{f}_{\text{conv}}(z) = \phi_s * \big(K * (\text{Ш}_s \odot z)\big) = K * \big(\phi_s * (\text{Ш}_s \odot z)\big) = K * z \qquad\qquad (2)$$

due to commutativity of convolution and the fact that discretization followed by convolution with ideal low-pass filter, both with same sampling rate $s$, is an identity operation, i.e., $\phi_s * (\text{Ш}_s \odot z) = z$. In other words, the convolution operates by continuously sliding the discretized kernel over the continuous representation of the feature map. This convolution introduces no new frequencies, so the bandlimit requirements for both translation and rotation equivariance are trivially fulfilled.

Convolution also commutes with translation in the continuous domain, and thus the operation is equivariant to translation. For rotation equivariance, the discrete kernel $K$ needs to be radially symmetric. We later show in Section 3.2 that trivially symmetric 1×1 convolution kernels are, despite their simplicity, a viable choice for rotation equivariant generative networks.

**Upsampling and downsampling**    Ideal upsampling does not modify the continuous representation. Its only purpose is to increase the output sampling rate ($s' > s$) to add headroom in the spectrum where subsequent layers may introduce additional content. Translation and rotation equivariance follow directly from upsampling being an identity operation in the continuous domain. With $\mathbf{f}_{\text{up}}(z) = z$, the discrete operation according to Eq. 1 is $\mathbf{F}_{\text{up}}(Z) = \text{Ш}_{s'} \odot (\phi_s * Z)$. If we choose $s' = ns$ with integer $n$, this operation can be implemented by first interleaving $Z$ with zeros to increase its sampling rate and then convolving it with a discretized filter $\text{Ш}_{s'} \odot \phi_s$.

In downsampling, we must low-pass filter $z$ to remove frequencies above the output bandlimit, so that the signal can be represented faithfully in the coarser discretization. The operation in continuous domain is $\mathbf{f}_{\text{down}}(z) = \psi_{s'} * z$, where an ideal low-pass filter $\psi_s := s^2 \cdot \phi_s$ is simply the corresponding interpolation filter normalized to unit mass. The discrete counterpart is $\mathbf{F}_{\text{down}}(Z) = \text{Ш}_{s'} \odot \big(\psi_{s'} * (\phi_s * Z)\big) = 1/s^2 \cdot \text{Ш}_{s'} \odot (\psi_{s'} * \psi_s * Z) = (s'/s)^2 \cdot \text{Ш}_{s'} \odot (\phi_{s'} * Z)$. The latter equality follows from $\psi_s * \psi_{s'} = \psi_{\min(s,s')}$. Similar to upsampling, downsampling by an integer fraction can be implemented with a discrete convolution followed by dropping sample points. Translation equivariance follows automatically from the commutativity of $\mathbf{f}_{\text{down}}(z)$ with translation, but for rotation equivariance we must replace $\phi_{s'}$ with a radially symmetric filter with disc-shaped frequency response. The ideal such filter [9] is given by $\phi_s^{\circ}(\boldsymbol{x}) = \text{jinc}(s\|\boldsymbol{x}\|) = 2J_1(\pi s\|\boldsymbol{x}\|)/(\pi s\|\boldsymbol{x}\|)$, where $J_1$ is the first order Bessel function of the first kind.

| | Configuration | FID↓ | EQ-T↑ | EQ-R↑ | | Parameter | FID↓ | EQ-T↑ | EQ-R↑ | Time | Mem. |
|---|---|---|---|---|---|---|---|---|---|---|---|
| A | StyleGAN2 | 5.14 | – | – | | Filter size $n = 4$ | 4.72 | 57.49 | 39.70 | **0.84×** | **0.99×** |
| B | + Fourier features | 4.79 | 16.23 | 10.81 | * | Filter size $n = 6$ | **4.50** | **66.65** | 40.48 | 1.00× | 1.00× |
| C | + No noise inputs | 4.54 | 15.81 | 10.84 | | Filter size $n = 8$ | 4.66 | 65.57 | **42.09** | 1.18× | 1.01× |
| D | + Simplified generator | 5.21 | 19.47 | 10.41 | | Upsampling $m = 1$ | **4.38** | 39.96 | 36.42 | **0.65×** | **0.87×** |
| E | + Boundaries & upsampling | 6.02 | 24.62 | 10.97 | * | Upsampling $m = 2$ | 4.50 | 66.65 | 40.48 | 1.00× | 1.00× |
| F | + Filtered nonlinearities | 6.35 | 30.60 | 10.81 | | Upsampling $m = 4$ | 4.57 | **74.21** | **40.97** | 2.31× | 1.62× |
| G | + Non-critical sampling | 4.78 | 43.90 | 10.84 | | Stopband $f_{t,0} = 2^{1.5}$ | 4.62 | 51.10 | 29.14 | **0.86×** | **0.90×** |
| H | + Transformed Fourier features | 4.64 | 45.20 | 10.61 | * | Stopband $f_{t,0} = 2^{2.1}$ | **4.50** | 66.65 | 40.48 | 1.00× | 1.00× |
| T | + Flexible layers  (StyleGAN3-T) | 4.62 | 63.01 | 13.12 | | Stopband $f_{t,0} = 2^{3.1}$ | 4.68 | **73.13** | **41.63** | 1.36× | 1.25× |
| R | + Rotation equiv. (StyleGAN3-R) | **4.50** | **66.65** | **40.48** | | | | | | | |

Figure 3: Results for FFHQ-U (unaligned FFHQ) at $256^2$. **Left:** Training configurations. FID is computed between 50k generated images and all training images [23, 28]; lower is better. EQ-T and EQ-R are our equivariance metrics in decibels (dB); higher is better. **Right:** Parameter ablations using our final configuration (R) for the filter's support, magnification around nonlinearities, and the minimum stopband frequency at the first layer. * indicates our default choices.

**Nonlinearity**   Applying a pointwise nonlinearity $\sigma$ in the discrete domain does not commute with fractional translation or rotation. However, in the continuous domain, any pointwise function commutes trivially with geometric transformations and is thus equivariant to translation and rotation. Fulfilling the bandlimit constraint is another question — applying, e.g., ReLU in the continuous domain may introduce arbitrarily high frequencies that cannot be represented in the output.

A natural solution is to eliminate the offending high-frequency content by convolving the continuous result with the ideal low-pass filter $\psi_s$. Then, the continuous representation of the nonlinearity becomes $\mathbf{f}_\sigma(z) = \psi_s * \sigma(z) = s^2 \cdot \phi_s * \sigma(z)$ and the discrete counterpart is $\mathbf{F}_\sigma(Z) = s^2 \cdot \text{III}_s \odot (\phi_s * \sigma(\phi_s * Z))$ (see Figure 2, right). This discrete operation cannot be realized without temporarily entering the continuous representation. We approximate this by upsampling the signal, applying the nonlinearity in the higher resolution, and downsampling it afterwards. Even though the nonlinearity is still performed in the discrete domain, we have found that only a $2\times$ temporary resolution increase is sufficient for high-quality equivariance. For rotation equivariance, we must use the radially symmetric interpolation filter $\phi_s^\circ$ in the downsampling step, as discussed above.

Note that nonlinearity is the only operation capable of generating novel frequencies in our formulation, and that we can limit the range of these novel frequencies by applying a reconstruction filter with a lower cutoff than $s/2$ before the final discretization operation. This gives us precise control over how much new information is introduced by each layer of a generator network (Section 3.2).

## 3   Practical application to generator network

We will now apply the theoretical ideas from the previous section in practice, by converting the well-established StyleGAN2 [30] generator to be fully equivariant to translation and rotation. We will introduce the necessary changes step-by-step, evaluating their impact in Figure 3. The discriminator remains unchanged in our experiments.

The StyleGAN2 generator consists of two parts. First, a *mapping network* transforms an initial, normally distributed latent to an intermediate latent code $\mathbf{w} \sim \mathcal{W}$. Then, a *synthesis network* $\mathbf{G}$ starts from a learned $4\times4\times512$ constant $Z_0$ and applies a sequence of $N$ layers — consisting of convolutions, nonlinearities, upsampling, and per-pixel noise — to produce an output image $Z_N = \mathbf{G}(Z_0; \mathbf{w})$. The intermediate latent code $\mathbf{w}$ controls the modulation of the convolution kernels in $\mathbf{G}$. The layers follow a rigid $2\times$ upsampling schedule, where two layers are executed at each resolution and the number of feature maps is halved after each upsampling. Additionally, StyleGAN2 employs skip connections, mixing regularization [29], and path length regularization.

Our goal is to make every layer of $\mathbf{G}$ equivariant w.r.t. the continuous signal, so that all finer details transform together with the coarser features of a local neighborhood. If this succeeds, the entire network becomes similarly equivariant. In other words, we aim to make the *continuous* operation $\mathbf{g}$ of the synthesis network equivariant w.r.t. transformations $\mathbf{t}$ (translations and rotations) applied on the continuous input $z_0$: $\mathbf{g}(\mathbf{t}[z_0]; \mathbf{w}) = \mathbf{t}[\mathbf{g}(z_0; \mathbf{w})]$. To evaluate the impact of various architectural changes and practical approximations, we need a way to measure how well the network implements the equivariances. For translation equivariance, we report the peak signal-to-noise ratio (PSNR) in decibels (dB) between two sets of images, obtained by translating the input and output of the

synthesis network by a random amount, resembling the definition by Zhang [61]:

$$\text{EQ-T} = 10 \cdot \log_{10} \left( I_{max}^2 / \mathbb{E}_{\mathbf{w}\sim\mathcal{W},x\sim\mathcal{X}^2,p\sim\mathcal{V},c\sim\mathcal{C}} \left[ \left( \mathbf{g}(\mathbf{t}_x[z_0];\mathbf{w})_c(p) - \mathbf{t}_x[\mathbf{g}(z_0;\mathbf{w})]_c(p) \right)^2 \right] \right) \quad (3)$$

Each pair of images, corresponding to a different random choice of $\mathbf{w}$, is sampled at integer pixel locations $p$ within their mutually valid region $\mathcal{V}$. Color channels $c$ are processed independently, and the intended dynamic range of generated images $-1 \ldots +1$ gives $I_{max} = 2$. Operator $\mathbf{t}_x$ implements spatial translation with 2D offset $x$, here drawn from distribution $\mathcal{X}^2$ of integer offsets. We define an analogous metric EQ-R for rotations, with the rotation angles drawn from $\mathcal{U}(0°, 360°)$. Appendix E in the Supplement gives implementation details and our accompanying videos highlight the practical relevance of different dB values.

### 3.1 Fourier features and baseline simplifications (configs B–D)

To facilitate exact continuous translation and rotation of the input $z_0$, we replace the learned input constant in StyleGAN2 with Fourier features [48, 58], which also has the advantage of naturally defining a spatially infinite map. We sample the frequencies uniformly within the circular frequency band $f_c = 2$, matching the original $4\times4$ input resolution, and keep them fixed over the course of training. This change (configs A and B in Figure 3, left) slightly improves FID and, crucially, allows us to compute the equivariance metrics without having to approximate the operator $\mathbf{t}$. This baseline architecture is far from being equivariant; our accompanying videos show that the output images deteriorate drastically when the input features are translated or rotated from their original position.

Next, we remove the per-pixel noise inputs because they are strongly at odds with our goal of a natural transformation hierarchy, i.e., that the exact sub-pixel position of each feature is exclusively inherited from the underlying coarse features. While this change (config C) is approximately FID-neutral, it fails to improve the equivariance metrics when considered in isolation.

To further simplify the setup, we decrease the mapping network depth as recommended by Karras et al. [28] and disable mixing regularization and path length regularization [30]. Finally, we also eliminate the output skip connections. We hypothesize that their benefit is mostly related to gradient magnitude dynamics during training and address the underlying issue more directly using a simple normalization before each convolution. We track the exponential moving average $\sigma^2 = \mathbb{E}[x^2]$ over all pixels and feature maps during training, and divide the feature maps by $\sqrt{\sigma^2}$. In practice, we bake the division into the convolution weights to improve efficiency. These changes (config D) bring FID back to the level of original StyleGAN2, while leading to a slight improvement in translation equivariance.

### 3.2 Step-by-step redesign motivated by continuous interpretation

**Boundaries and upsampling (config E)** Our theory assumes an infinite spatial extent for the feature maps, which we approximate by maintaining a fixed-size margin around the target canvas, cropping to this extended canvas after each layer. This explicit extension is necessary as border padding is known to leak absolute image coordinates into the internal representations [25, 31, 58]. In practice, we have found a 10-pixel margin to be enough; further increase has no noticeable effect on the results.

Motivated by our theoretical model, we replace the bilinear $2\times$ upsampling filter with a better approximation of the ideal low-pass filter. We use a windowed $\text{sinc}$ filter with a relatively large Kaiser window [35] of size $n = 6$, meaning that each output pixel is affected by 6 input pixels in upsampling and each input pixel affects 6 output pixels in downsampling. Kaiser window is a particularly good choice for our purposes, because it offers explicit control over the transition band and attenuation (Figure 4a). In the remainder of this section, we specify the transition band explicitly and compute the remaining parameters using Kaiser's original formulas (Appendix C). For now, we choose to employ *critical sampling* and set the filter cutoff $f_c = s/2$, i.e., exactly at the bandlimit, and transition band half-width $f_h = (\sqrt{2} - 1)(s/2)$. Recall that sampling rate $s$ equals the width of the canvas in pixels, given our definitions in Section 2.

The improved handling of boundaries and upsampling (config E) leads to better translation equivariance. However, FID is compromised by 16%, probably because we started to constrain what the feature maps can contain. In a further ablation (Figure 3, right), smaller resampling filters ($n = 4$) hurt translation equivariance, while larger filters ($n = 8$) mainly increase training time.

**Filtered nonlinearities (config F)** Our theoretical treatment of nonlinearities calls for wrapping each leaky ReLU (or any other commonly used non-linearity) between $m\times$ upsampling and $m\times$

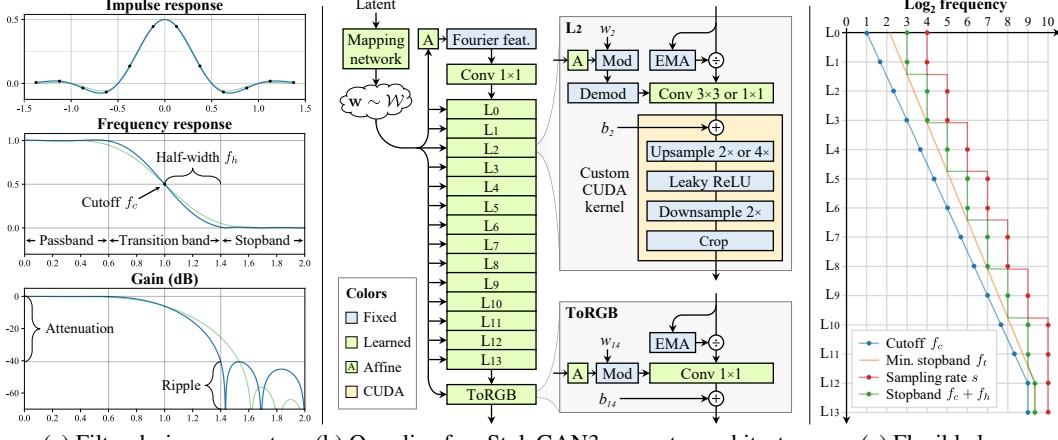

(a) Filter design concepts     (b) Our alias-free StyleGAN3 generator architecture     (c) Flexible layers

Figure 4: **(a)** 1D example of a $2\times$ upsampling filter with $n = 6$, $s = 2$, $f_c = 1$, and $f_h = 0.4$ (blue). Setting $f_h = 0.6$ makes the transition band wider (green), which reduces the unwanted stopband ripple and thus leads to stronger attenuation. **(b)** Our alias-free generator, corresponding to configs T and R in Figure 3. The main datapath consists of Fourier features and normalization (Section 3.1), modulated convolutions [30], and filtered nonlinearities (Section 3.2). **(c)** Flexible layer specifications (config T) with $N = 14$ and $s_N = 1024$. Cutoff $f_c$ (blue) and minimum acceptable stopband frequency $f_t$ (orange) obey geometric progression over the layers; sampling rate $s$ (red) and actual stopband $f_c + f_h$ (green) are computed according to our design constraints.

downsampling, for some magnification factor $m$. We further note that the order of upsampling and convolution can be switched by virtue of the signal being bandlimited, allowing us to fuse the regular $2\times$ upsampling and a subsequent $m\times$ upsampling related to the nonlinearity into a single $2m\times$ upsampling. In practice, we find $m = 2$ to be sufficient (Figure 3, right), again improving EQ-T (config F). Implementing the upsample-LReLU-downsample sequence is not efficient using the primitives available in current deep learning frameworks [1, 39], and thus we implement a custom CUDA kernel (Appendix D) that combines these operations (Figure 4b), leading to $10\times$ faster training and considerable memory savings.

**Non-critical sampling (config G)**   The critical sampling scheme—where filter cutoff is set exactly at the bandlimit—is ideal for many image processing applications as it strikes a good balance between antialiasing and the retention of high-frequency detail [50]. However, our goals are markedly different because aliasing is highly detrimental for the equivariance of the generator. While high-frequency detail is important in the output image and thus in the highest-resolution layers, it is less important in the earlier ones given that their exact resolutions are somewhat arbitrary to begin with.

To suppress aliasing, we can simply lower the cutoff frequency to $f_c = s/2 - f_h$, which ensures that all alias frequencies (above $s/2$) are in the stopband.[3] For example, lowering the cutoff of the blue filter in Figure 4a would move its frequency response left so that the the worst-case attenuation of alias frequencies improves from $6\,\text{dB}$ to $40\,\text{dB}$. This *oversampling* can be seen as a computational cost of better antialiasing, as we now use the same number of samples to express a slower-varying signal than before. In practice, we choose to lower $f_c$ on all layers except the highest-resolution ones, because in the end the generator must be able to produce crisp images to match the training data. As the signals now contain less spatial information, we modify the heuristic used for determining the number of feature maps to be inversely proportional to $f_c$ instead of the sampling rate $s$. These changes (config G) further improve translation equivariance and push FID below the original StyleGAN2.

**Transformed Fourier features (config H)**   Equivariant generator layers are well suited for modeling unaligned and arbitrarily oriented datasets, because any geometric transformation introduced to the intermediate features $z_i$ will directly carry over to the final image $z_N$. Due to the limited capability of the layers themselves to introduce global transformations, however, the input features $z_0$ play a crucial role in defining the global orientation of $z_N$. To let the orientation vary on a per-image basis,

---

[3]Here, $f_c$ and $f_h$ correspond to the output (downsampling) filter of each layer. The input (upsampling) filters are based on the properties of the incoming signal, i.e., the output filter parameters of the previous layer.

the generator should have the ability to transform $z_0$ based on $\mathbf{w}$. This motivates us to introduce a learned affine layer that outputs global translation and rotation parameters for the input Fourier features (Figure 4b and Appendix F). The layer is initialized to perform an identity transformation, but learns to use the mechanism over time when beneficial; in config H this improves the FID slightly.

**Flexible layer specifications (config T)**   Our changes have improved the equivariance quality considerably, but some visible artifacts still remain as our accompanying videos demonstrate. On closer inspection, it turns out that the attenuation of our filters (as defined for config G) is still insufficient for the lowest-resolution layers. These layers tend to have rich frequency content near their bandlimit, which calls for extremely strong attenuation to completely eliminate aliasing.

So far, we have used the rigid sampling rate progression from StyleGAN2, coupled with simplistic choices for filter cutoff $f_c$ and half-width $f_h$, but this need not be the case; we are free to specialize these parameters on a per-layer basis. In particular, we would like $f_h$ to be high in the lowest-resolution layers to maximize attenuation in the stopband, but low in the highest-resolution layers to allow matching high-frequency details of the training data.

Figure 4c illustrates an example progression of filter parameters in a 14-layer generator with two critically sampled full-resolution layers at the end. The cutoff frequency grows geometrically from $f_c = 2$ in the first layer to $f_c = s_N/2$ in the first critically sampled layer. We choose the minimum acceptable stopband frequency to start at $f_{t,0} = 2^{2.1}$, and it grows geometrically but slower than the cutoff frequency. In our tests, the stopband target at the last layer is $f_t = f_c \cdot 2^{0.3}$, but the progression is halted at the first critically sampled layer. Next, we set the sampling rate $s$ for each layer so that it accommodates frequencies up to $f_t$, rounding up to the next power of two without exceeding the output resolution. Finally, to maximize the attenuation of aliasing frequencies, we set the transition band half-width to $f_h = \max(s/2, f_t) - f_c$, i.e., making it as wide as possible within the limits of the sampling rate, but at least wide enough to reach $f_t$. The resulting improvement depends on how much slack is left between $f_t$ and $s/2$; as an extreme example, the first layer stopband attenuation improves from $42\,\mathrm{dB}$ to $480\,\mathrm{dB}$ using this scheme.

The new layer specifications again improve translation equivariance (config T), eliminating the remaining artifacts. A further ablation (Figure 3, right) shows that $f_{t,0}$ provides an effective way to trade training speed for equivariance quality. Note that the number of layers is now a free parameter that does not directly depend on the output resolution. In fact, we have found that a fixed choice of $N$ works consistently across multiple output resolutions and makes other hyperparameters such as learning rate behave more predictably. We use $N = 14$ in the remainder of this paper.

**Rotation equivariance (config R)**   We obtain a rotation equivariant version of the network with two changes. First, we replace the $3\times3$ convolutions with $1\times1$ on all layers and compensate for the reduced capacity by doubling the number of feature maps. Only the upsampling and downsampling operations spread information between pixels in this config. Second, we replace the sinc-based downsampling filter with a radially symmetric jinc-based one that we construct using the same Kaiser scheme (Appendix C). We do this for all layers except the two critically sampled ones, where it is important to match the potentially non-radial spectrum of the training data. These changes (config R) improve EQ-R without harming FID, even though each layer has 56% fewer trainable parameters.

We also employ an additional stabilization trick in this configuration. Early on in the training, we blur all images the discriminator sees using a Gaussian filter. We start with $\sigma = 10$ pixels, which we ramp to zero over the first 200k images. This prevents the discriminator from focusing too heavily on high frequencies early on. Without this trick, config R is prone to early collapses because the generator sometimes learns to produce high frequencies with a small delay, trivializing the discriminator's task.

## 4   Results

Figure 5 gives results for six datasets using StyleGAN2 [30] as well as our alias-free StyleGAN3-T and StyleGAN3-R generators. In addition to the standard FFHQ [29] and METFACES [28], we created unaligned versions of them. We also created a properly resampled version of AFHQ [14] and collected a new BEACHES dataset. Appendix B describes the datasets in detail. The results show that our FID remains competitive with StyleGAN2. StyleGAN3-T and StyleGAN3-R perform equally well in terms of FID, and both show a very high level of translation equivariance. As expected, only the latter provides rotation equivariance. In FFHQ (1024×1024) the three generators had 30.0M, 22.3M and 15.8M parameters, while the training times were 1106, 1576 (+42%) and 2248

| Dataset | Config | FID↓ | EQ-T↑ | EQ-R↑ |
|---|---|---|---|---|
| FFHQ-U
70000 img, 1024² 
Train from scratch | StyleGAN2 | 3.79 | 15.89 | 10.79 |
| | StyleGAN3-T (ours) | 3.67 | 61.69 | 13.95 |
| | StyleGAN3-R (ours) | **3.66** | 64.78 | 47.64 |
| FFHQ
70000 img, 1024² 
Train from scratch | StyleGAN2 | **2.70** | 13.58 | 10.22 |
| | StyleGAN3-T (ours) | 2.79 | 61.21 | 13.82 |
| | StyleGAN3-R (ours) | 3.07 | 64.76 | 46.62 |
| METFACES-U
1336 img, 1024² 
ADA, from FFHQ-U | StyleGAN2 | 18.98 | 18.77 | 13.19 |
| | StyleGAN3-T (ours) | **18.75** | 64.11 | 16.63 |
| | StyleGAN3-R (ours) | **18.75** | 66.34 | 48.57 |
| METFACES
1336 img, 1024² 
ADA, from FFHQ | StyleGAN2 | 15.22 | 16.39 | 12.89 |
| | StyleGAN3-T (ours) | **15.11** | 65.23 | 16.82 |
| | StyleGAN3-R (ours) | 15.33 | 64.86 | 46.81 |
| AFHQv2
15803 img, 512² 
ADA, from scratch | StyleGAN2 | 4.62 | 13.83 | 11.50 |
| | StyleGAN3-T (ours) | **4.04** | 60.15 | 13.51 |
| | StyleGAN3-R (ours) | 4.40 | 64.89 | 40.34 |
| BEACHES
20155 img, 512² 
ADA, from scratch | StyleGAN2 | 5.03 | 15.73 | 12.69 |
| | StyleGAN3-T (ours) | **4.32** | 59.33 | 15.88 |
| | StyleGAN3-R (ours) | 4.57 | **63.66** | 37.42 |

| Ablation | Translation eq. | | + Rotation eq. | | |
|---|---|---|---|---|---|
| | FID↓ | EQ-T↑ | FID↓ | EQ-T↑ | EQ-R↑ |
| * Main configuration | 4.62 | 63.01 | **4.50** | 66.65 | 40.48 |
| With mixing reg. | **4.60** | 63.48 | 4.67 | 63.59 | 40.90 |
| With noise inputs | 4.96 | 24.46 | 5.79 | 26.71 | 26.80 |
| Without flexible layers | 4.64 | 45.20 | 4.65 | 44.74 | 22.52 |
| Fixed Fourier features | 5.93 | 64.57 | 6.48 | 66.20 | 41.77 |
| With path length reg. | 5.00 | **68.36** | 5.98 | **71.64** | **42.18** |
| 0.5× capacity | 7.43 | 63.14 | 6.52 | 63.08 | 39.89 |
| * 1.0× capacity | 4.62 | 63.01 | 4.50 | 66.65 | 40.48 |
| 2.0× capacity | **3.80** | **66.61** | **4.18** | 70.06 | 42.51 |
| * Kaiser filter, $n = 6$ | 4.62 | 63.01 | 4.50 | 66.65 | 40.48 |
| Lanczos filter, $a = 2$ | 4.69 | 51.93 | **4.44** | 57.70 | 25.25 |
| Gaussian filter, $\sigma = 0.4$ | 5.91 | 56.89 | 5.73 | 59.53 | 39.43 |

| G-CNN comparison | FID↓ | EQ-T↑ | EQ-R↑ | Params | Time |
|---|---|---|---|---|---|
| * StyleGAN3-T (ours) | 4.62 | 63.01 | 13.12 | 23.3M | **1.00×** |
| + $p4$ symmetry [16] | 4.69 | 61.90 | 17.07 | 21.8M | 2.48× |
| * StyleGAN3-R (ours) | **4.50** | **66.65** | **40.48** | 15.8M | 1.37× |

Figure 5: **Left:** Results for six datasets. We use adaptive discriminator augmentation (ADA) [28] for the smaller datasets. "StyleGAN2" corresponds to our baseline config B with Fourier features. **Right:** Ablations and comparisons for FFHQ-U (unaligned FFHQ) at 256². * indicates our default choices.

(+103%) GPU hours. Our accompanying videos show side-by-side comparisons with StyleGAN2, demonstrating visually that the texture sticking problem has been solved. The resulting motion is much more natural, better sustaining an illusion that there is a coherent 3D scene being imaged.

**Ablations and comparisons**    In Section 3.1 we disabled a number of StyleGAN2 features. We can now turn them on one by one to gauge their effect on our generators (Figure 5, right). While mixing regularization can be re-enabled without any ill effects, we also find that styles can be mixed quite reliably even without this explicit regularization (Appendix A). Re-enabling noise inputs or relying on StyleGAN2's original layer specifications compromises equivariances significantly, and using fixed Fourier features or re-enabling path length regularization harms FID. Path length regularization is in principle at odds with translation equivariance, as it penalizes image changes upon latent space walk and thus encourages texture sticking. We suspect that the counterintuitive improvement in equivariance may come from slightly blurrier generated images, at a cost of poor FID.

In a scaling test we tried changing the number of feature maps, observing that equivariances remain at a high level, but FID suffers considerably when the capacity is halved. Doubling the capacity improves result quality in terms of FID, at the cost of almost 4× training time. Finally, we consider alternatives for our windowed Kaiser filter. Lanczos is competitive in terms of FID, but as a separable filter it compromises rotation equivariance in particular. Gaussian leads to clearly worse FIDs.

We compare StyleGAN3-R to an alternative where the rotation part is implemented using $p4$ symmetric G-CNN [16, 17] on top of our StyleGAN3-T. This approach provides only modest rotation equivariance while being slower to train. Steerable filters [55] could theoretically provide competitive EQ-R, but the memory and training time requirements proved infeasible with generator networks of this size.

Appendix A demonstrates that the spectral properties of generated images closely match training data, comparing favorably to several earlier architectures.

**Internal representations**    Figure 6 visualizes typical internal representations from the networks. While in StyleGAN2 all feature maps seem to encode signal magnitudes, in our networks some of the maps take a different role and encode phase information instead. Clearly this is something that is needed when the network synthesizes detail *on the surfaces*; it needs to invent a coordinate system. In StyleGAN3-R, the emergent positional encoding patterns appear to be somewhat more well-defined. We believe that the existence of a coordinate system that allows precise localization on the surfaces of objects will prove useful in various applications, including advanced image and video editing.

## 5   Limitations, discussion, and future work

In this work we modified only the generator, but it seems likely that further benefits would be available by making the discriminator equivariant as well. For example, in our FFHQ results the teeth do not move correctly when the head turns, and we suspect that this is caused by the discriminator

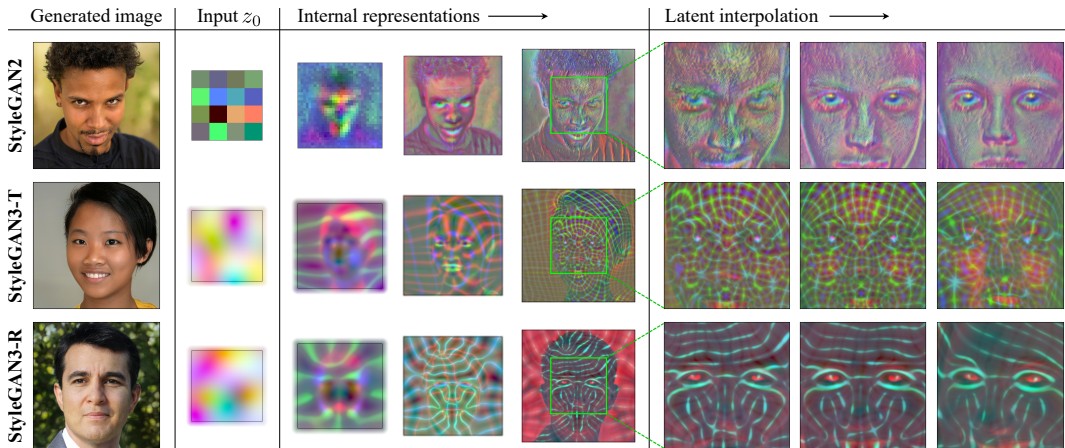

Figure 6: Example internal representations (3 feature maps as RGB) in StyleGAN2 and our generators.

accidentally preferring to see the front teeth at certain pixel locations. Concurrent work has identified that aliasing is detrimental for such generalization [51].

Our alias-free generator architecture contains implicit assumptions about the nature of the training data, and violating these may cause training difficulties. Let us consider an example. Suppose we have black-and-white cartoons as training data that we (incorrectly) pre-process using point sampling [38], leading to training images where almost all pixels are either black or white and the edges are jagged. This kind of badly aliased training data is difficult for GANs in general, but it is especially at odds with equivariance: on the one hand, we are asking the generator to be able to translate the output smoothly by subpixel amounts, but on the other hand, edges must still remain jagged and pixels only black/white, to remain faithful to the training data. The same issue can also arise with letterboxing of training images, low-quality JPEGs, or retro pixel graphics, where the jagged stair-step edges are a defining feature of the aesthetic. In such cases it may be beneficial for the generator to be aware of the pixel grid.

In future, it might be interesting to re-introduce noise inputs (stochastic variation) in a way that is consistent with hierarchical synthesis. A better path length regularization would encourage neighboring features to move together, not discourage them from moving at all. It might be beneficial to try to extend our approach to equivariance w.r.t. scaling, anisotropic scaling, or even arbitrary homeomorphisms. Finally, it is well known that antialiasing should be done before tone mapping. So far, all GANs — including ours — have operated in the sRGB color space (after tone mapping).

Attention layers in the middle of a generator [60] could likely be dealt with similarly to non-linearities by temporarily switching to higher resolution – although the time complexity of attention layers may make this somewhat challenging in practice. Recent attention-based GANs that start with a tokenizing transformer (e.g., VQGAN [18]) may be at odds with equivariance. Whether it is possible to make them equivariant is an important open question.

**Potential negative societal impacts** of (image-producing) GANs include many forms of disinformation, from fake portraits in social media [24] to propaganda videos of world leaders [43]. Our contribution eliminates certain characteristic artifacts from videos, potentially making them more convincing or deceiving, depending on the application. Viable solutions include model watermarking [59] along with large-scale authenticity assessment in major social media sites. This entire project consumed 92 GPU years and 225 MWh of electricity on an in-house cluster of NVIDIA V100s. The new StyleGAN3 generator is only marginally costlier to train or use than that of StyleGAN2.

## 6 Acknowledgments

We thank David Luebke, Ming-Yu Liu, Koki Nagano, Tuomas Kynkäänniemi, and Timo Viitanen for reviewing early drafts and helpful suggestions. Frédo Durand for early discussions. Tero Kuosmanen for maintaining our compute infrastructure. AFHQ authors for an updated version of their dataset. Getty Images for the training images in the BEACHES dataset. We did not receive external funding or additional revenues for this project.

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
