# Supplemental Material:
# Alias-Free Generative Adversarial Networks

## A  Additional results

Uncurated sets of samples for StyleGAN2 (baseline config B with Fourier features) and our alias-free generators StyleGAN3-T and StyleGAN3-R are shown in Figures 7 (FFHQ-U), 8 (METFACES-U), 9 (AFHQV2), and 10 (BEACHES). Truncation trick was not used when generating the images.

StyleGAN2 and our generators yield comparable FIDs in all of these datasets. Visual inspection did not reveal anything surprising in the first three datasets, but in BEACHES our new generators seem to generate a somewhat reduced set of possible scene layouts properly. We suspect that this is related to the lack of noise inputs, which forces the generators to waste capacity for what is essentially random number generation [16]. Finding a way to reintroduce noise inputs without breaking equivariances is therefore an important avenue of future work.

The accompanying interpolation videos reveal major differences between StyleGAN2 and StyleGAN3-R. For example, in METFACES much of details such as brushstrokes or cracked paint seems to be glued to the pixel coordinates in StyleGAN2, whereas with StyleGAN3 all details move together with the depicted model. The same is evident in AFHQV2 with the fur moving credibly in StyleGAN3 interpolations, while mostly sticking to the image coordinates in StyleGAN2. In BEACHES we furthermore observe that StyleGAN2 tends to "fade in" details while retaining a mostly fixed viewing position, while StyleGAN3 creates plenty of apparent rotations and movement. The videos use hand-picked seeds to better showcase the relevant effects.

In a further test we created two example cinemagraphs that mimic small-scale head movement and facial animation in FFHQ. The geometric head motion was generated as a random latent space walk along hand-picked directions from GANSpace [10] and SeFa [22]. The changes in expression were realized by applying the "global directions" method of StyleCLIP [21], using the prompts "angry face", "laughing face", "kissing face", "sad face", "singing face", and "surprised face". The differences between StyleGAN2 and StyleGAN3 are again very prominent, with the former displaying jarring sticking of facial hair and skin texture, even under subtle movements.

The equivariance quality videos illustrate the practical relevance of the PSNR numbers in Figures 3 and 5 of the main paper. We observe that for EQ-T numbers over ∼50 dB indicate high-quality results, and for EQ-R ∼40 dB look good.

We also provide an animated version of the nonlinearity visualization in Figure 2.

In style mixing [16] two or more independently chosen latent codes are fed into different layers of the generator. Ideally all combinations would produce images that are not obviously broken, and furthermore, it would be desirable that specific layers end up controlling well-defined semantic aspects in the images. StyleGAN uses mixing regularization [16] during training to achieve these goals. We observe that mixing regularization continues to work similarly in StyleGAN3, but we also wanted to know whether it is truly necessary because the regularization is known to be detrimental for many complex and multi-modal datasets [9]. When we disable the regularization, obviously broken images remain rare, based on a visual inspection of a large number of images. The semantically meaningful controls are somewhat compromised, however, as Figure 11 shows.

Figure 12 compares the convergence of our main configurations (config T and R) against the results of Karras et al. [16, 14]. The overall shape of the curves is similar; introducing translation and rotation equivariance in the generator does not appear to significantly alter the training dynamics.

Following recent works that address signal processing issues in GANs [1, 6], we show average power spectra of the generated and real images in Figure 13. The plots are computed from images that

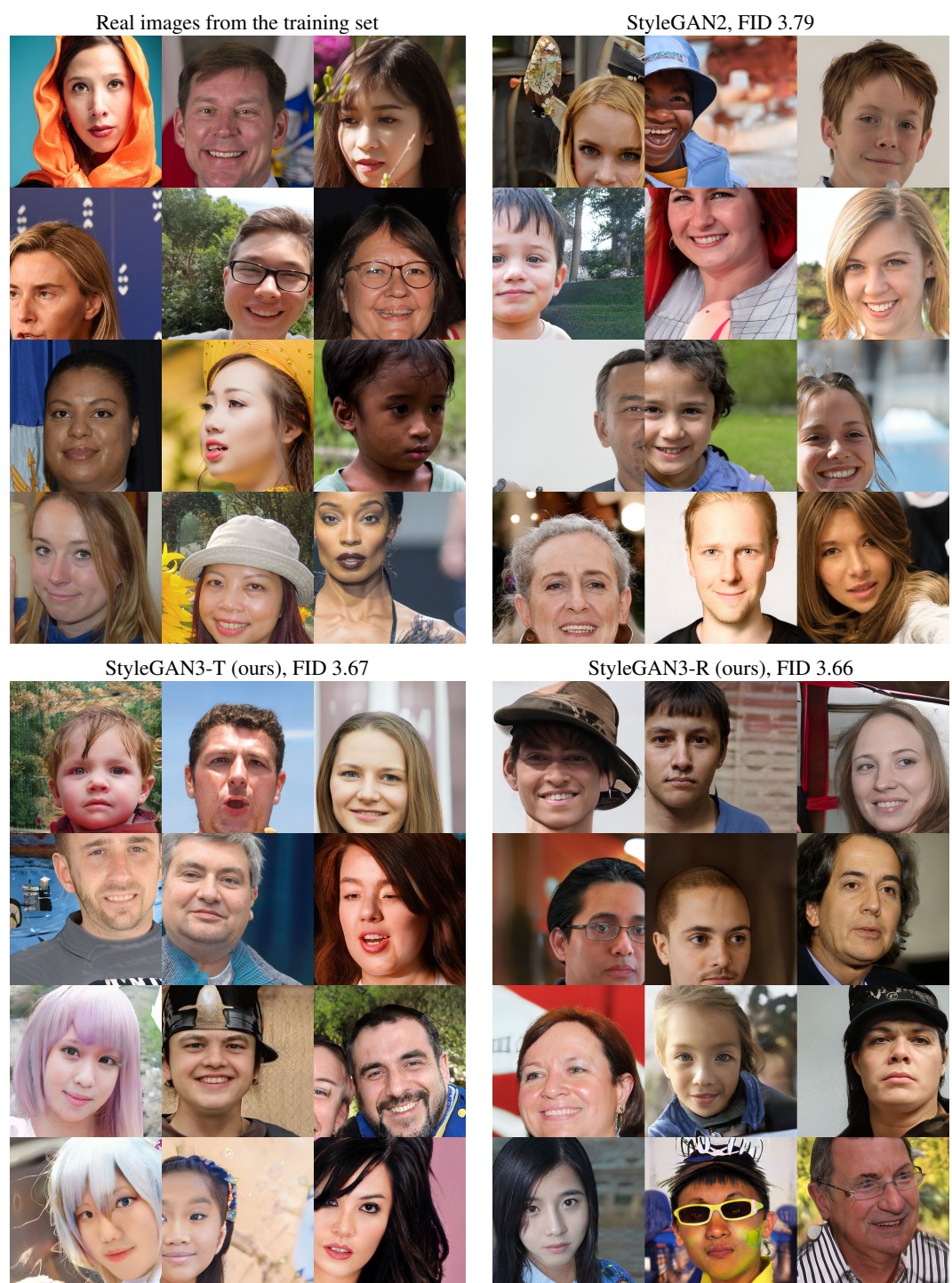

Figure 7: Uncurated samples for unaligned FFHQ (FFHQ-U). Truncation was not used.

Real images from the training set StyleGAN2, FID 18.98

StyleGAN3-T (ours), FID 18.75 StyleGAN3-R (ours), FID 18.75

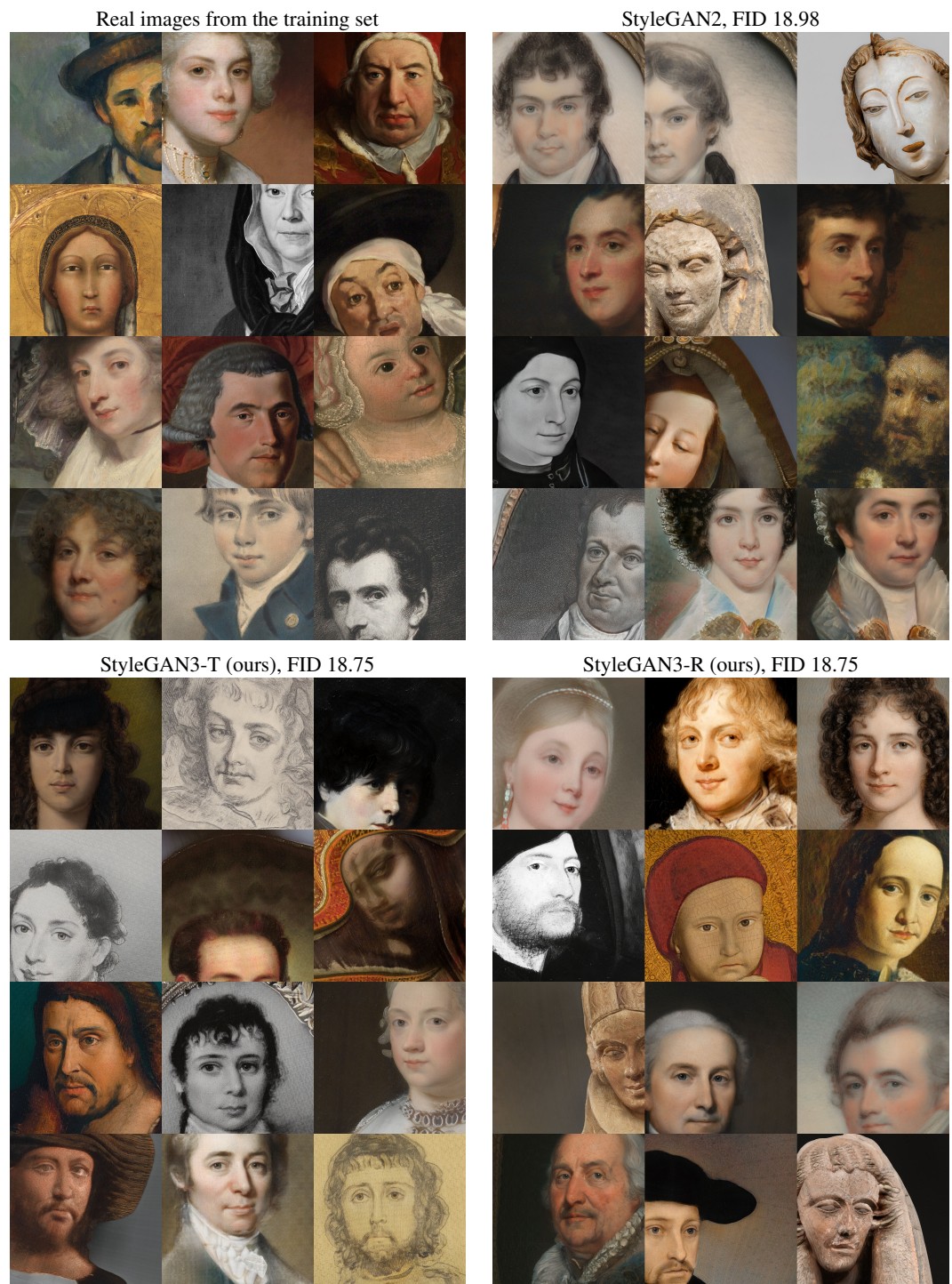

Figure 8: Uncurated samples for unaligned MetFaces (Metfaces-U). Truncation was not used.

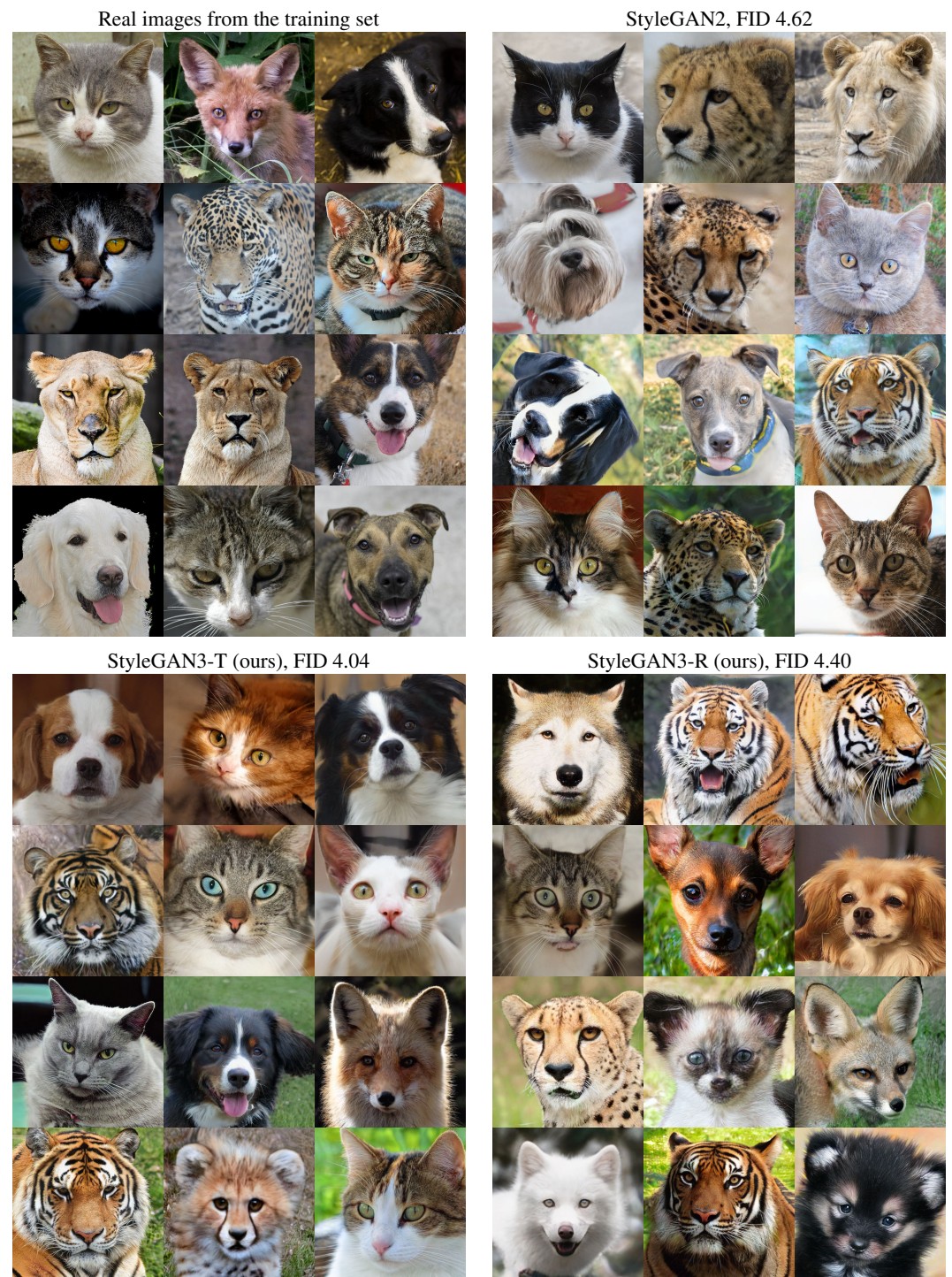

Figure 9: Uncurated samples for AFHQV2. Truncation was not used.

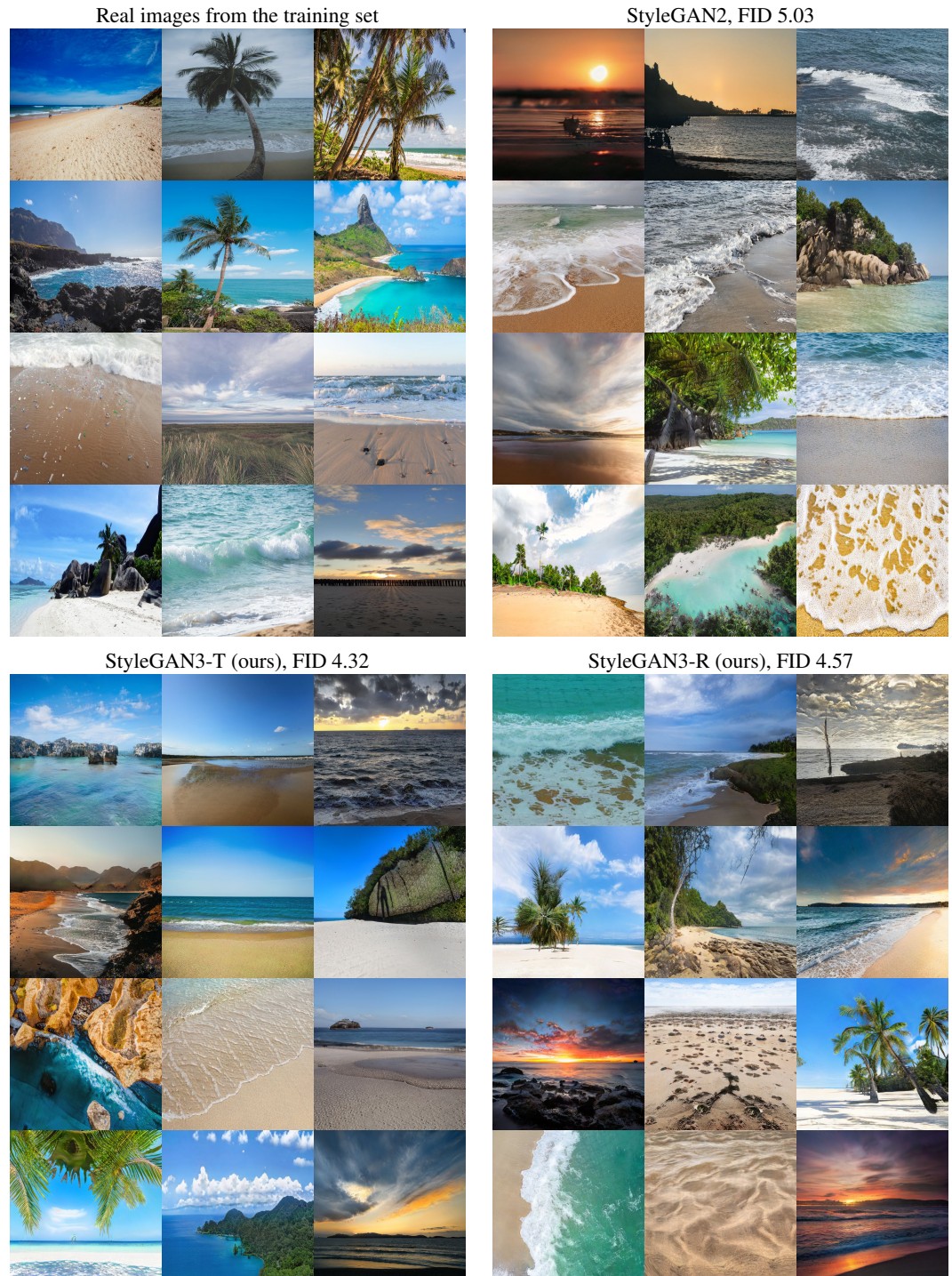

Figure 10: Uncurated samples for BEACHES. Truncation was not used.

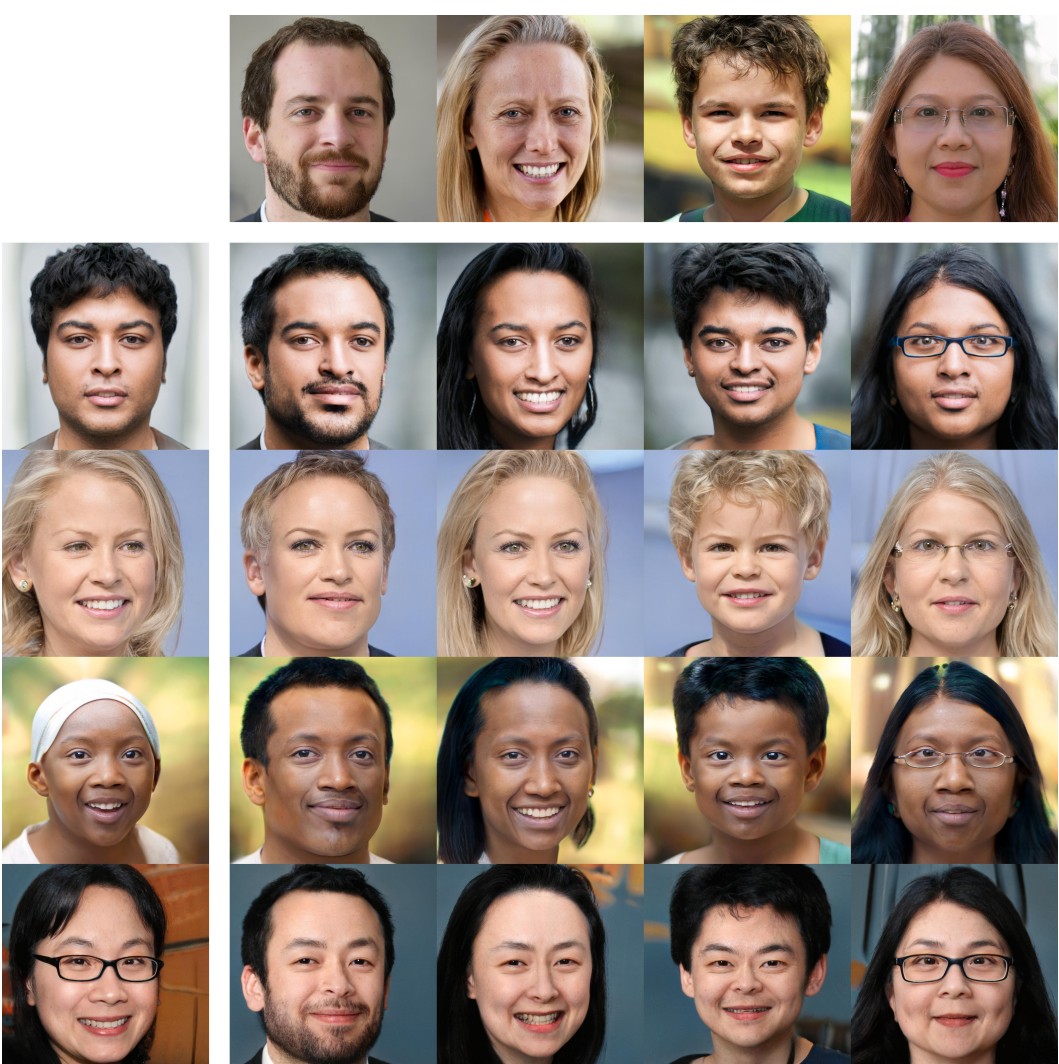

Figure 11: Hand-picked style mixing examples where the coarse (0–6) and fine (7–14) layers use a different latent code. Mixing regularization was not used during training. Head pose, coarse facial shape, hair length and glasses seem to get inherited from the coarse layers (top row), while coloring and finer facial features are mostly inherited from the fine layers (leftmost column). The control is not quite perfect: e.g., feminine/masculine features are not reliably copied from exactly one of the sources. Moving the fine/coarse boundary fixes this particular issue, but other similar problems persist.

are whitened with the overall training dataset mean and standard deviation. Because FFT interprets the signal as periodic, we eliminate the sharp step edge across the image borders by windowing the pixel values prior to the transform. This eliminates the axis-aligned cross artifact which may obscure meaningful detail in the spectrum. We display the average 2D spectrum as a contour plot, which makes the orientation-dependent falloff apparent, and highlights detail like regularly spaced residuals of upsampling grids, and fixed noise patterns. We also plot 1D slices of the spectrum along the horizontal and diagonal angle without azimuthal integration, so as to not average out the detail. The code for reproducing these steps is included in the public release.

# B  Datasets

In this section, we describe the new datasets and list the licenses of all datasets.

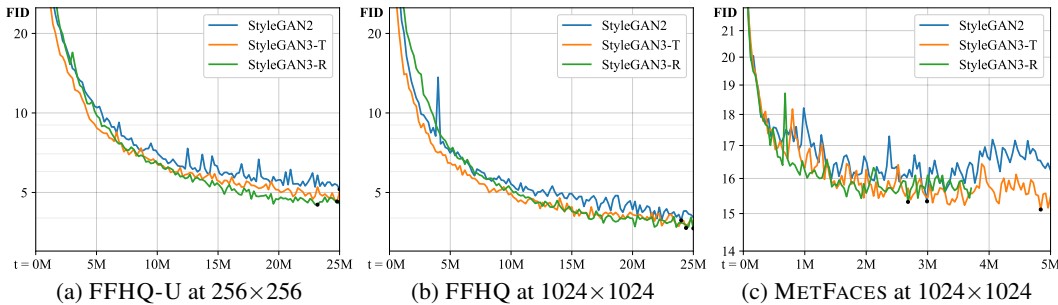

(a) FFHQ-U at 256×256     (b) FFHQ at 1024×1024     (c) METFACES at 1024×1024

Figure 12: Training convergence with three datasets using StyleGAN2 and our main configurations (config T and R). $x$-axis corresponds to the total number of real images shown to the discriminator and $y$-axis is the Fréchet inception distance (FID), computed between 50k generated images and all training images [11, 14]; lower is better. The black dots indicate the best FID for each training run, matching the corresponding cases in Figures 3 and 5. METFACES was trained using adaptive discriminator augmentation (ADA) [14], starting from the corresponding FFHQ snapshot with the lowest FID.

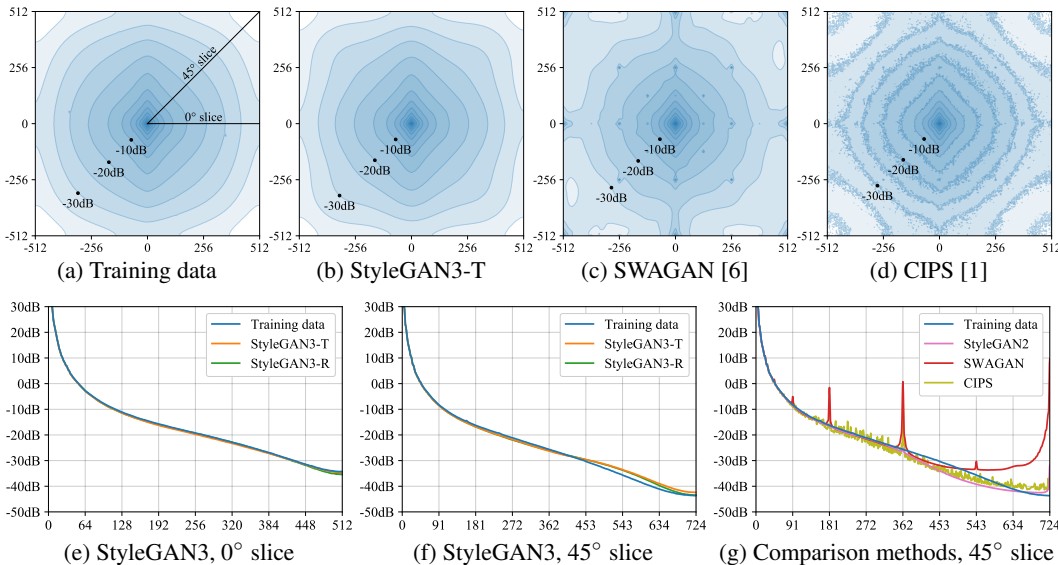

(a) Training data    (b) StyleGAN3-T    (c) SWAGAN [6]    (d) CIPS [1]

(e) StyleGAN3, 0° slice    (f) StyleGAN3, 45° slice    (g) Comparison methods, 45° slice

Figure 13: **Top:** Average 2D power spectrum of the training images in FFHQ at 1024×1024 resolution, along with the corresponding spectra of random images generated using StyleGAN3-T, SWAGAN [6], and CIPS [1]. Each plot represents the average power over 70k images, computed as follows. From each image, we subtract the training dataset mean, after which we divide it by the training dataset standard deviation. Note that these normalizing quantities represent the entire dataset reduced to two scalars, and do not vary by color channel or pixel coordinate. The image is then multiplied with a separable Kaiser window with $\beta = 8$, and its power spectrum is computed as the absolute values of the FFT raised to the second power. This processing is applied to each color channel separately and the result is averaged over them. These spectra are then averaged over all the images in the dataset. The result is plotted on the decibel scale. **Bottom:** One-dimensional slices of the power spectra at 0° and 45° angles.

## B.1 FFHQ-U and MetFaces-U

We built unaligned variants of the existing FFHQ [15] and METFACES [14] datasets. The originals are available at `https://github.com/NVlabs/ffhq-dataset` and `https://github.com/NVlabs/metfaces-dataset`, respectively. The datasets were rebuilt with a modification of the original procedure based on the original code, raw uncropped images, and facial landmark metadata. The code required to reproduce the modified datasets is included in the public release.

We use axis-aligned crop rectangles, and do not rotate them to match the orientation of the face. This retains the natural variation of camera and head tilt angles. Note that the images are still generally upright, i.e., never upside down or at $90°$ angle. The scale of the rectangle is determined as before. For each image, the crop rectangle is randomly shifted from its original face-centered position, with the horizontal and vertical offset independently drawn from a normal distribution. The standard deviation is chosen as $20\%$ of the crop rectangle dimension. If the crop rectangle falls partially outside the original image boundaries, we keep drawing new random offsets until we find one that does not. This removes the need to pad the images with fictional mirrored content, and we explicitly disabled this feature of the original build script.

Aside from the exact image content, the number of images and other specifications match the original dataset exactly. While FFHQ-U contains identifiable images of persons, it does not introduce new images beyond those already in the original FFHQ.

### B.2 AFHQv2

We used an updated version of the AFHQ dataset [3] where the resampling filtering has been improved. The original dataset suffers from pixel-level artifacts caused by inadequate downsampling filters [20]. This caused convergence problems with our models, as the sharp "stair-step" aliasing artifacts are difficult to reproduce without direct access to the pixel grid.

The dataset was rebuilt using the original uncropped images and crop rectangle metadata, using the PIL library implementation of Lanczos resampling as recommended by Parmar et al. [20]. In a minority of cases, the crop rectangles were modified to remove non-isotropic scaling and other unnecessary transformations. A small amount ($\sim 2\%$) of images were dropped for technical reasons, leaving a total of $15803$ images. Aside from this, the specifications of the dataset match the original. We use all images of all the three classes (cats, dogs, and wild animals) as one training dataset.

### B.3 Beaches

BEACHES is a new dataset of $20155$ photographs of beaches at resolution $512\times512$. The training images were provided by Getty Images. BEACHES is a proprietary dataset that we are licensed to use, but not to redistribute. We are therefore unable to release the full training data or pre-trained models for this dataset.

### B.4 Licenses

The FFHQ dataset is available under Creative Commons BY-NC-SA 4.0 license by NVIDIA Corporation, and consist of images published by respective authors under Creative Commons BY 2.0, Creative Commons BY-NC 2.0, Public Domain Mark 1.0, Public Domain CC0 1.0, and U.S. Government Works license.

The METFACES dataset is available under Creative Commons BY-NC 2.0 license by NVIDIA Corporation, and consists of images available under the Creative Commons Zero (CC0) license by the Metropolitan Museum of Art.

The original AFHQ dataset is available at `https://github.com/clovaai/stargan-v2` under Creative Commons BY-NC 4.0 license by NAVER Corporation.

## C   Filter details

In this section, we review basic FIR filter design methodology and detail the recipe used to construct the upsampling and downsampling filters in our generator. We start with simple Kaiser filters in one dimension, discussing parameter selection and the necessary modifications needed for upsampling and downsampling. We then proceed to extend the filters to two dimensions and conclude by detailing the alternative filters evaluated in Figure 5, right. Our definitions are consistent with standard signal processing literature (e.g., Oppenheim [19]) as well as widely used software packages (e.g., `scipy.signal.firwin`).

## C.1 Kaiser low-pass filters

In one dimension, the ideal continuous-time low-pass filter with cutoff $f_c$ is given by $\psi(x) = 2f_c \cdot \text{sinc}(2f_c x)$, where $\text{sinc}(x) = \sin(\pi x)/(\pi x)$. The ideal filter has infinite attenuation in the stopband, i.e., it completely eliminates all frequencies above $f_c$. However, its impulse response is also infinite, which makes it impractical for three reasons: implementation efficiency, border artifacts, and *ringing* caused by long-distance interactions. The most common way to overcome these issues is to limit the spatial extent of the filter using the window method [19]:

$$h_K(x) = 2f_c \cdot \text{sinc}(2f_c x) \cdot w_K(x), \tag{4}$$

where $w_K(x)$ is a *window function* and $h_K(x)$ is the resulting practical approximation of $\psi(x)$. Different window functions represent different tradeoffs between the frequency response and spatial extent; the smaller the spatial extent, the weaker the attenuation. In this paper we use the Kaiser window [12], also known as the Kaiser–Bessel window, that provides explicit control over this tradeoff. The Kaiser window is defined as

$$w_K(x) = \begin{cases} I_0\left(\beta\sqrt{1 - (2x/L)^2}\right)\big/I_0(\beta), & \text{if } |x| \leq L/2, \\ 0, & \text{if } |x| > L/2, \end{cases} \tag{5}$$

where $L$ is the desired spatial extent, $\beta$ is a free parameter that controls the shape of the window, and $I_0$ is the zeroth-order modified Bessel function of the first kind. Note that the window has discontinuities at $\pm L/2$; the value is strictly positive at $x = L/2$ but zero at $x = L/2 + \epsilon$.

When operating on discretely sampled signals, it is necessary to discretize the filter as well:

$$h_K[i] = h_K\left((i - (n-1)/2)/s\right)/s, \quad \text{for } i \in \{0, 1, \ldots, n-1\}, \tag{6}$$

where $h_K[i]$ is the discretized version of $h_K(x)$ and $s$ is the sampling rate. The filter is defined at $n$ discrete spatial locations, i.e., *taps*, located $1/s$ units apart and placed symmetrically around zero. Given the values of $n$ and $s$, the spatial extent can be expressed as $L = (n-1)/s$. An odd value of $n$ results in a *zero-phase* filter that preserves the original sample locations, whereas an even value shifts the sample locations by $1/(2s)$ units.

The filters considered in this paper are approximately normalized by construction, i.e., $\int_x h_K(x) \approx \sum_i h_K[i] \approx 1$. Nevertheless, we have found it beneficial to explicitly normalize them after discretization. In other words, we strictly enforce $\sum_i h_K[i] = 1$ by scaling the filter taps to reduce the risk of introducing cumulative scaling errors when the signal is passed through several consecutive layers.

## C.2 Selecting window parameters

Kaiser [12] provides convenient empirical formulas to connect the parameters of $w_K$ to the properties of $h_K$. Given the number of taps and the desired transition band width, the maximum attenuation achievable with $h_K[i]$ is approximated by

$$A = 2.285 \cdot (n-1) \cdot \pi \cdot \Delta f + 7.95, \tag{7}$$

where $A$ is the attenuation measured in decibels and $\Delta f$ is the width of the transition band expressed as a fraction of $s/2$. We choose to define the transition band using half-width $f_h$, which gives $\Delta f = (2f_h)/(s/2)$. Given the value of $A$, the optimal choice for the shape parameter $\beta$ is then approximated [12] by

$$\beta = \begin{cases} 0.1102 \cdot (A - 8.7), & \text{if } A > 50, \\ 0.5842 \cdot (A - 21)^{0.4} + 0.07886 \cdot (A - 21), & \text{if } 21 \leq A \leq 50, \\ 0, & \text{if } A < 21, \end{cases} \tag{8}$$

This leaves us with two free parameters: $n$ controls the spatial extent while $f_h$ controls the transition band. The choice of these parameters directly influences the resulting attenuation; increasing either parameter yields a higher value for $A$.

## C.3 Upsampling and downsampling

When upsampling a signal, i.e., $\mathbf{F}_{\text{up}}(Z) = \text{III}_{s'} \odot (\phi_s * Z) = 1/s^2 \cdot \text{III}_{s'} \odot (\psi_s * Z)$, we are concerned not only the with input sampling rate $s$, but also with the output sampling rate $s'$. With an integer upsampling factor $m$, we can think of the upsampling operation as consisting of two steps: we first increase the sampling rate to $s' = s \cdot m$ by interleaving $m-1$ zeros between each input sample by and then low-pass filter the resulting signal to eliminate the alias frequencies above $s/2$. In order to keep the signal magnitude unchanged, we must also scale the result by $m$ with one-dimensional signals, or by $m^2$ with two-dimensional signals. Since the filter now operates under $s'$ instead of $s$, we must adjust its parameters accordingly:

$$n' = n \cdot m, \qquad L' = (n'-1)/s', \qquad \Delta f' = (2f_h)/(s'/2), \qquad (9)$$

which gives us the final upsampling filter

$$h'_K[i] = h'_K\Big(\big(i - (n'-1)/2\big)/s'\Big)/s', \quad \text{for } i \in \{0, 1, \ldots, n'-1\}. \qquad (10)$$

Multiplying the number of taps by $m$ keeps the spatial extent of the filter unchanged with respect to the input samples, and it also compensates for the reduced attenuation from $\Delta f' < \Delta f$. Note that if the upsampling factor is even, $n'$ will be even as well, meaning that $h'_K$ shifts the sample locations by $1/(2s')$. This is the desired behavior — if we consider sample $i$ to represent the continuous interval $[i \cdot s, (i+1) \cdot s]$ in the input signal, the same interval will be represented by $m$ consecutive samples $m \cdot i, \ldots, m \cdot i + m - 1$ in the output signal. Using a zero-phase upsampling filter, i.e., an odd value for $n'$, would break this symmetry, leading to inconsistent behavior with respect to the boundaries. Note that our symmetric interpretation is common in many computer graphics APIs, such as OpenGL, and it is also reflected in our definition of the Dirac comb III in Section 2.

Upsampling and downsampling are adjoint operations with respect to each other, disregarding the scaling of the signal magnitude. This means that the above definitions are readily applicable to downsampling as well; to downsample a signal by factor $m$, we first filter it by $h'_K$ and then discard the last $m-1$ samples within each group of $m$ consecutive samples. The interpretation of all filter parameters, as well as the sample locations, is analogous to the upsampling case.

## C.4 Two-dimensional filters

Any one-dimensional filter, including $h_K$, can be trivially extended to two dimensions by defining the corresponding separable filter

$$h_K^+(\boldsymbol{x}) = h_K(x_0) \cdot h_K(x_1) = (2f_c)^2 \cdot \text{sinc}(2f_c x_0) \cdot \text{sinc}(2f_c x_1) \cdot w_K(x_0) \cdot w_K(x_1), \qquad (11)$$

where $\boldsymbol{x} = (x_0, x_1)$. $h_K^+$ has the same cutoff as $h_K$ along the coordinate axes, i.e., $\boldsymbol{f}_{c,x} = (f_c, 0)$ and $\boldsymbol{f}_{c,y} = (0, f_c)$, and its frequency response forms a square shape over the 2D plane, implying that the cutoff frequency along the diagonal is $\boldsymbol{f}_{c,d} = (f_c, f_c)$. In practice, a separable filter can be implemented efficiently by first filtering each row of the two-dimensional signal independently with $h_K$ and then doing the same for each column. This makes $h_K^+$ an ideal choice for all upsampling filters in our generator, as well as the downsampling filters in configs A–T (Figure 3, left).

The fact that the spectrum of $h_K^+$ is not radially symmetric, i.e., $\|\boldsymbol{f}_{c,d}\| \neq \|\boldsymbol{f}_{c,x}\|$, is problematic considering config R. If we rotate the input feature maps of a given layer, their frequency content will rotate as well. To enforce rotation equivariant behavior, we must ensure that the effective cutoff frequencies remain unchanged by this. The ideal radially symmetric low-pass filter [2] is given by $\psi_s^{\circ}(\boldsymbol{x}) = (2f_c)^2 \cdot \text{jinc}(2f_c\|\boldsymbol{x}\|)$. The jinc function, also known as besinc, sombrero function, or Airy disk, is defined as $\text{jinc}(x) = 2J_1(\pi x)/(\pi x)$, where $J_1$ is the first order Bessel function of the first kind. Using the same windowing scheme as before, we define the corresponding practical filter as

$$h_K^{\circ}(\boldsymbol{x}) = (2f_c)^2 \cdot \text{jinc}(2f_c\|\boldsymbol{x}\|) \cdot w_K(x_0) \cdot w_K(x_1). \qquad (12)$$

Note that even though jinc is radially symmetric, we still treat the window function as separable in order to retain its spectral properties. In config R, we perform all downsampling operations using $h_K^{\circ}$, except for the last two critically sampled layers where we revert to $h_K^+$.

## C.5 Alternative filters

In Figure 5, right, we compare the effectiveness of Kaiser filters against two alternatives: Lanczos and Gaussian. These filters are typically defined using prototypical filter kernels $k_L$ and $k_G$, respectively:

$$k_L(x) = \begin{cases} \mathrm{sinc}(x) \cdot \mathrm{sinc}(x/a), & \text{if } |x| < a, \\ 0, & \text{if } |x| \geq a, \end{cases} \tag{13}$$

$$k_G(x) = \exp\left(-\frac{1}{2}(x/\sigma)^2\right) \Big/ \left(\sigma\sqrt{2\pi}\right), \tag{14}$$

where $a$ is the spatial extent of the Lanczos kernel, typically set to 2 or 3, and $\sigma$ is the standard deviation of the Gaussian kernel. In Figure 5 of the main paper we set $a = 2$ and $\sigma = 0.4$; we tested several different values and found these choices to work reasonably well.

The main shortcoming of the prototypical kernels is that they do not provide an explicit way to control the cutoff frequency. In order to enable apples-to-apples comparison, we assume that the kernels have an implicit cutoff frequency at 0.5 and scale their impulse responses to account for the varying $f_c$:

$$h_L(x) = 2f_c \cdot k_L(2f_c x), \qquad\qquad h_G(x) = 2f_c \cdot k_G(2f_c x). \tag{15}$$

We limit the computational complexity of the Gaussian filter by enforcing $h_G(x) = 0$ when $|x| > 8/s$, with respect to the input sampling rate in the upsampling case. In practice, $h_G(x)$ is already very close to zero in this range, so the effect of this approximation is negligible. Finally, we extend the filters to two dimensions by defining the corresponding separable filters:

$$h_L^+(\boldsymbol{x}) = (2f_c)^2 \cdot k_L(2f_c x_0) \cdot k_L(2f_c x_1), \quad h_G^+(\boldsymbol{x}) = (2f_c)^2 \cdot k_G(2f_c x_0) \cdot k_G(2f_c x_1). \tag{16}$$

Note that $h_G^+$ is radially symmetric by construction, which makes it ideal for rotation equivariance. $h_L^+$, however, has no widely accepted radially symmetric counterpart, so we simply use the same separable filter in config R as well.

# D   Custom CUDA kernel for filtered nonlinearity

Implementing the upsample-nonlinearity-downsample sequence is inefficient using the standard primitives available in modern deep learning frameworks. The intermediate feature maps have to be transferred between on-chip and off-chip GPU memory multiple times and retained for the backward pass. This is especially costly because the intermediate steps operate on upsampled, high-resolution data. To overcome this, we implement the entire sequence as a single operation using a custom CUDA kernel. This improves training performance by approximately an order of magnitude thanks to reduced memory traffic, and also decreases GPU memory usage significantly.

The combined kernel consists of four phases: input, upsampling, nonlinearity, and downsampling. The computation is parallelized by subdividing the output feature maps into non-overlapping tiles, and computing one output tile per CUDA thread block. First, in input phase, the corresponding input region is read into on-chip shared memory of the thread block. Note that the input regions for neighboring output tiles will overlap spatially due to the spatial extent of filters.

The execution of up-/downsampling phases depends on whether the corresponding filters are separable or not. For a separable filter, we perform vertical and horizontal 1D convolutions sequentially, whereas a non-separable filter requires a single 2D convolution. All these convolutions and the nonlinearity operate in on-chip shared memory, and only the final output of the downsampling phase is written to off-chip GPU memory.

## D.1   Gradient computation

To compute gradients of the combined operation, they need to propagate through each of the phases in reverse order. Fortunately, the combined upsample-nonlinearity-downsample operation is mostly self-adjoint with proper changes in parameters, e.g., swapping the up-/downsampling factors and the associated filters. The only problematic part is the nonlinearity that is performed in the upsampled resolution. A naïve but general solution would be to store the intermediate high-resolution input to the nonlinearity, but the memory consumption would be infeasible for training large models.

| | upsample 2× downsample 2× | | | | upsample 4× downsample 2× | | | | upsample 2× downsample 4× | | | |
|---|---|---|---|---|---|---|---|---|---|---|---|---|
| Sep. up | yes | yes | no | no | yes | yes | no | no | yes | yes | no | no |
| Sep. down | yes | no | yes | no | yes | no | yes | no | yes | no | yes | no |
| PyTorch (ms) | 7.88 | 12.40 | 12.68 | 17.12 | 10.07 | 31.51 | 14.96 | 36.33 | 39.35 | 56.73 | 125.83 | 143.15 |
| Ours (ms) | 0.42 | 0.59 | 0.66 | 0.92 | 0.49 | 0.84 | 0.80 | 1.01 | 1.20 | 1.89 | 3.04 | 3.66 |
| Speedup × | 19 | 21 | 19 | 19 | 21 | 38 | 19 | 36 | 33 | 30 | 41 | 39 |

Figure 14: Upsample-nonlinearity-downsample timings in milliseconds using native PyTorch operations vs our optimized CUDA kernel. The benchmarks were run on NVIDIA Titan V GPU, using input size $512 \times 512 \times 32$ and filter size $n = 6$, i.e., $n' = 12$ and $n' = 24$ for up-/downsampling rates of 2 and 4, respectively. Sep. up and Sep. down indicate the use of separable up-/downsampling filters.

Our kernel is specialized to use leaky ReLU as the nonlinearity, which offers a straightforward way to conserve memory: to propagate gradients, it is sufficient to know whether the corresponding input value to nonlinearity was positive or negative. When using 16-bit floating-point datatypes, there is an additional complication because the outputs of the nonlinearity need to be clamped [14], and when this occurs, the corresponding gradients must be zero. Therefore, in the forward pass we store two bits of auxiliary information per value to cover the three possible cases: positive, negative, or clamped. In the backward pass, reading these bits is sufficient for correct gradient computation — no other information from the forward pass is needed.

### D.2 Optimizations for common upsampling factors

Let us consider one-dimensional 2× upsampling where the input is (virtually) interleaved with zeros and convolved with an $n'$-tap filter where $n' = 2n$ (cf. Equation 9). There are $n$ nonzero input values under the $n'$-tap kernel, so if each output pixel is computed separately, the convolution requires $n$ multiply-add operations per pixel and equally many shared memory load instructions, for a total of $2n$ instructions per output pixel.[1] However, note that the computation of two neighboring output pixels accesses only $n + 1$ input pixels in total. By computing two output pixels at a time and avoiding redundant shared memory load instructions, we obtain an average cost of $\frac{3}{2}n + \frac{1}{2}$ instructions per pixel — close to 25% savings. For 4× upsampling, we can similarly reduce the instruction count by up to 37.5% by computing four output pixels at a time. We apply these optimizations in 2× and 4× upsampling for both separable and non-separable filters.

Figure 14 benchmarks the performance of our kernel with various up-/downsampling factors and with separable and non-separable filters. In network layers that keep the sampling rate fixed, both factors are 2×, whereas layers that increase the sampling rate by a factor of two, 4× upsampling is combined with 2× downsampling. The remaining combination of 2× upsampling and 4× downsampling is needed when computing gradients of the latter case. The speedup over native PyTorch operations varies between ∼20–40×, which yields an overall training speedup of approximately 10×.

## E  Equivariance metrics

In this section, we describe our equivariance metrics, EQ-T and EQ-R, in detail. We also present additional results using an alternative translation metric, EQ-T$_{\text{frac}}$, based on fractional sub-pixel translation.

We express each of our metrics as the *peak signal-to-noise ratio* (PSNR) between two sets of images, measured in decibels (dB). PSNR is a commonly used metric in image restoration literature. In the typical setting we have two signals, reference $I$ and its noisy approximation $K$, defined over discrete domain $\mathcal{D}$ — usually a two-dimensional pixel grid. The PSNR between $I$ and $K$ is then defined via the mean squared error (MSE):

$$\text{MSE}_{\mathcal{D}}(I, K) = \frac{1}{\|\mathcal{D}\|} \sum_{i \in \mathcal{D}} \left( I[i] - K[i] \right)^2, \tag{17}$$

$$\text{PSNR}_{\mathcal{D}}(I, K) = 10 \cdot \log_{10} \left( \frac{I_{max}^2}{\text{MSE}_{\mathcal{D}}(I, K)} \right), \tag{18}$$

---

[1]Input of the upsampling is stored in shared memory, but the filter weights can be stored in CUDA constant memory where they can be accessed without a separate load instruction.

| | Configuration | FID | EQ-T | EQ-T$_{\text{frac}}$ | | Parameter | FID | EQ-T | EQ-T$_{\text{frac}}$ |
|---|---|---|---|---|---|---|---|---|---|
| A | StyleGAN2 | 5.14 | – | – | | Filter size $n = 4$ | 4.72 | 57.49 | 44.65 |
| B | + Fourier features | 4.79 | 16.23 | 16.28 | * | Filter size $n = 6$ | **4.50** | **66.65** | 45.92 |
| C | + No noise inputs | 4.54 | 15.81 | 15.84 | | Filter size $n = 8$ | 4.66 | 65.57 | **46.57** |
| D | + Simplified generator | 5.21 | 19.47 | 19.57 | | Upsampling $m = 1$ | **4.38** | 39.96 | 37.55 |
| E | + Boundaries & upsampling | 6.02 | 24.62 | 24.70 | * | Upsampling $m = 2$ | 4.50 | 66.65 | 45.92 |
| F | + Filtered nonlinearities | 6.35 | 30.60 | 30.68 | | Upsampling $m = 4$ | 4.57 | **74.21** | **46.81** |
| G | + Non-critical sampling | 4.78 | 43.90 | 42.24 | | Stopband $f_{t,0} = 2^{1.5}$ | 4.62 | 51.10 | 44.46 |
| H | + Transformed Fourier features | 4.64 | 45.20 | 42.78 | * | Stopband $f_{t,0} = 2^{2.1}$ | **4.50** | 66.65 | 45.92 |
| T | + Flexible layers  (StyleGAN3-T) | 4.62 | 63.01 | **46.40** | | Stopband $f_{t,0} = 2^{3.1}$ | 4.68 | **73.13** | **46.27** |
| R | + Rotation equiv. (StyleGAN3-R) | **4.50** | **66.65** | 45.92 | | | | | |

Figure 15: Results with our alternative translation equivariance metric EQ-T$_{\text{frac}}$; higher is better.

where $\text{MSE}_{\mathcal{D}}(I, K)$ is the average squared difference between matching elements of $I$ and $K$. $I_{max}$ is the expected dynamic range of the reference signal, i.e., $I_{max} \approx \max_{i \in \mathcal{D}}(I[i]) - \min_{i \in \mathcal{D}}(I[i])$. The dynamic range is usually considered to be a global constant, e.g., the range of valid RGB values, as opposed to being dependent on the content of $I$. In our case, $I$ and $K$ represent desired and actual outputs of the synthesis network, respectively, with a dynamic range of $[-1, 1]$. This implies that $I_{max} = 2$. High PSNR values indicate that $K$ is close to $I$; in the extreme case, where $K = I$, we have $\text{PSNR}_{\mathcal{D}}(I, K) = \infty$ dB.

Since we are interested in *sets* of images, we use a slightly extended definition for MSE that allows $I$ and $K$ to be defined over an arbitrary, potentially uncountable domain:

$$\text{MSE}_{\mathcal{D}}(I, K) = \mathbb{E}_{i \sim \mathcal{D}} \left[ \left( I(i) - K(i) \right)^2 \right]. \tag{19}$$

### E.1 Integer translation

The goal of our integer translation metric, EQ-T, is to measure how closely, on average, the output the synthesis network $\mathbf{G}$ matches a translated reference image when we translate the input of $\mathbf{G}$. In other words,

$$\text{EQ-T} = \text{PSNR}_{\mathcal{W} \times \mathcal{X}^2 \times \mathcal{V} \times \mathcal{C}}(I_{\mathbf{t}}, K_{\mathbf{t}}),$$
$$I_{\mathbf{t}}(\mathbf{w}, \boldsymbol{x}, \boldsymbol{p}, c) = \mathbf{T}_{\boldsymbol{x}} \big[ \mathbf{G}(z_0; \mathbf{w}) \big][\boldsymbol{p}, c], \tag{20}$$
$$K_{\mathbf{t}}(\mathbf{w}, \boldsymbol{x}, \boldsymbol{p}, c) = \mathbf{G}(\mathbf{t}_{\boldsymbol{x}}[z_0]; \mathbf{w})[\boldsymbol{p}, c],$$

where $\mathbf{w} \sim \mathcal{W}$ is a random intermediate latent code produced by the mapping network, $\boldsymbol{x} = (x_0, x_1) \sim \mathcal{X}^2$ is a random translation offset, $\boldsymbol{p}$ enumerates pixel locations in the mutually valid region $\mathcal{V}$, $c \sim \mathcal{C}$ is the color channel, and $z_0$ represents the input Fourier features. For integer translations, we sample the translation offsets $x_0$ and $x_1$ from $\mathcal{X} = \mathcal{U}[-s_N/8, s_N/8]$, where $s_N$ is the width of the image in pixels.

In practice, we estimate the expectation in Equation 20 as an average over 50,000 random samples of $(\mathbf{w}, \boldsymbol{x}) \sim \mathcal{W} \times \mathcal{X}^2$. For given $\mathbf{w}$ and $\boldsymbol{x}$, we generate the reference image $I_{\mathbf{t}}$ by running the synthesis network and translating the resulting image by $\boldsymbol{x}$ pixels (operator $\mathbf{T}_{\boldsymbol{x}}$). We then obtain the approximate result image $K_{\mathbf{t}}$ by translating the input Fourier features by the corresponding amount (operator $\mathbf{t}_{\boldsymbol{x}}$), as discussed in Appendix F.1, and running the synthesis network again. The mutually valid region of $I_{\mathbf{t}}$ (translated by $(x_0, x_1)$) and $K_{\mathbf{t}}$ (translated by $(0, 0)$) is given by

$$\mathcal{V} = \{\max(x_0, 0), \ldots, s_N + \min(x_0, 0) - 1\} \times$$
$$\{\max(x_1, 0), \ldots, s_N + \min(x_1, 0) - 1\}. \tag{21}$$

### E.2 Fractional translation

Our translation equivariance metric has the nice property that, for a perfectly equivariant generator, the value of EQ-T converges to $\infty$ dB when the number of samples tends to infinity. However, this comes at the cost of completely ignoring subpixel effects. In fact, it is easy to imagine a generator that is perfectly equivariant to integer translation but fails with subpixel translation; in principle, this is true for any generator whose output is not properly bandlimited, including, e.g., implicit coordinate-based MLPs [1].

To verify that our generators *are* able to handle subpixel translation, we define an alternative translation equivariance metric, EQ-$\mathrm{T}_{\mathrm{frac}}$, where the translation offsets $x_0$ and $x_1$ are sampled from a continuous distribution $\mathcal{X} = \mathcal{U}(-s_N/8, s_N/8)$. While the continuous operator $\mathbf{t}_{\boldsymbol{x}}$ readily supports this new definition with fractional offsets, extending the discrete $\mathbf{T}_{\boldsymbol{x}}$ is slightly more tricky.

In practice, we define $\mathbf{T}_{\boldsymbol{x}}$ via standard Lanczos resampling, by filtering the image produced by $\mathbf{G}$ using the prototypical Lanczos filter (Equation 15) with $a = 3$, evaluated at integer tap locations offset by $\boldsymbol{x}$. We explicitly normalize the resulting discretized filter to enforce the partition of unity property. We also shrink the mutually valid region to account for the spatial extent $a$ by redefining

$$\mathcal{V} = \{\max(x_0 + a, 0), \ldots, s_N + \min(x_0 - a, -1)\} \times$$
$$\{\max(x_1 + a, 0), \ldots, s_N + \min(x_1 - a, -1)\}. \tag{22}$$

Figure 15 compares the results of the two metrics, EQ-T and EQ-$\mathrm{T}_{\mathrm{frac}}$, using the same training configurations as Figure 3 in the main paper. The metrics agree reasonably well up until $\sim$40 dB, after which the fractional metric starts to saturate; it consistently fails to rise above 50 dB in our tests. This is due to the fact that the definition of subpixel translation is inherently ambiguous. The choice of the resampling filter represents a tradeoff between aliasing, ringing, and retention of high frequencies; there is no reason to assume that the generator would necessarily have to make the same tradeoff as the metric. Based on the results, we conclude that our configs G–R are essentially perfectly equivariant to subpixel translation within the limits of Lanczos resampling's accuracy. However, due to its inherent limitations, we refrain from choosing EQ-$\mathrm{T}_{\mathrm{frac}}$ as our primary metric.

### E.3   Rotation

Measuring equivariance with respect to arbitrary rotations has the same fundamental limitation as our EQ-$\mathrm{T}_{\mathrm{frac}}$ metric: the resampling operation is inherently ambiguous, so we cannot except the results to be perfectly accurate beyond $\sim$40 dB. Arbitrary rotations also have the additional complication that the bandlimit of a discretely sampled image is not radially symmetric.

Consider rotating the continuous representation of a discretely sampled image by $45°$. The original frequency content of the image is constrained within the rectangular bandlimit $\boldsymbol{f} \in [-s_N/2, +s_N/2]^2$. The frequency content of the rotated image, however, forms a diamond shape that extends all the way to $\|\boldsymbol{f}\| = \sqrt{2}s_N/2$ along the main axes but only to $\|\boldsymbol{f}\| = s_N/2$ along the diagonals. In other words, it simultaneously has too much frequency content, but also too little. This has two implications. First, in order to obtain a valid discretized result image, we have to low-pass filter the image *both* before *and* after the rotation to completely eliminate aliasing. Second, even if we are successful in eliminating the aliasing, the rotated image will still lack the highest representable diagonal frequencies. The second point further implies that when computing PSNR, our reference image $I$ will inevitably lack some frequencies that are present in the output of $\mathbf{G}$. To obtain the correct result, we must eliminate these extraneous frequencies — without modifying the output image in any other way.

Based on the above reasoning, we define our EQ-R metric as follows:

$$\mathrm{EQ\text{-}R} = \mathrm{PSNR}_{\mathcal{W} \times \mathcal{A} \times \mathcal{V} \times \mathcal{C}}(I_{\mathbf{r}}, K_{\mathbf{r}}),$$
$$I_{\mathbf{r}}(\mathbf{w}, \alpha, \boldsymbol{p}, c) = \mathbf{R}_\alpha\big[\mathbf{G}(z_0; \mathbf{w})\big][\boldsymbol{p}, c], \tag{23}$$
$$K_{\mathbf{r}}(\mathbf{w}, \alpha, \boldsymbol{p}, c) = \mathbf{R}_\alpha^*\big[\mathbf{G}(\mathbf{r}_\alpha[z_0]; \mathbf{w})\big][\boldsymbol{p}, c],$$

where the random rotation angle $\alpha$ is drawn from $\mathcal{A} = \mathcal{U}(0°, 360°)$ and operator $\mathbf{r}_\alpha$ corresponds to continuous rotation of the input Fourier features by $\alpha$ with respect to the center of the canvas $[0, 1]^2$. $\mathbf{R}_\alpha$ corresponds to high-quality rotation of the reference image, and $\mathbf{R}_\alpha^*$ represents a *pseudo-rotation* operator that modifies the frequency content of the image *as if* it had undergone $\mathbf{R}_\alpha$ — but without *actually* rotating it.

The *ideal* rotation operator $\hat{\mathbf{R}}$ is easily defined under our theoretical framework presented in Section 2.1:
$$\hat{\mathbf{R}}_\alpha[Z] = \mathrm{III} \odot \big(\psi * \mathbf{r}_\alpha[\phi * Z]\big) = 1/s^2 \cdot \mathrm{III} \odot \big(\psi * \mathbf{r}_\alpha[\psi * Z]\big). \tag{24}$$
In other words, we first convolve the discretely sampled input image $Z$ with $\phi$ to obtain the corresponding continuous representation. We then rotate this continuous representation using $\mathbf{r}_\alpha$, bandlimit the result by convolving with $\psi$, and finally extract the corresponding discrete representation by

multiplying with Ⅲ. To reduce notational clutter, we omit the subscripts denoting the sampling rate $s$. We can swap the order of the rotation and a convolution in the above formula by rotating the kernel in the opposite direction to compensate:

$$\hat{\mathbf{R}}_\alpha[Z] = 1/s^2 \cdot \text{Ⅲ} \odot \mathbf{r}_\alpha[\hat{h}_R * Z], \qquad\qquad \hat{h}_R = \mathbf{r}_{-\alpha}[\psi] * \psi, \qquad (25)$$

where $\hat{h}_R$ represents an ideal "rotation filter" that bandlimits the signal with respect to both the input and the output. Its spectrum is the eight-sided polygonal intersection of the original and the rotated rectangle.

In order to obtain a practical approximation $\mathbf{R}_\alpha$, we must replace $\hat{h}_R$ with an approximate filter $h_R$ that has finite support. Given such a filter, we get $\mathbf{R}_\alpha[Z] = 1/s^2 \cdot \text{Ⅲ} \odot \mathbf{r}_\alpha[h_R * Z]$. In practice, we implement this operation using two additional approximations. First, we approximate $1/s^2 \cdot h_R * Z$ by an upsampling operation to a higher temporary resolution, using $h_R$ as the upsampling filter and $m = 4$. Second, we approximate $\text{Ⅲ} \odot \mathbf{r}_\alpha$ by performing a set of bilinear lookups from the temporary high-resolution image.

To obtain $h_R$, we again utilize the standard Lanczos window with $a = 3$:

$$h_R = \left(\mathbf{r}_{-\alpha}[\psi] * \psi\right) \odot \left(\mathbf{r}_{-\alpha}[w_L^+] * w_L^+\right), \qquad (26)$$

where we apply the same rotation-convolution to both the filter and the window function. $w_L^+$ corresponds the canonical separable Lanczos window, similar to the one used in Equation 15:

$$w_L^+(\boldsymbol{x}) = \begin{cases} \text{sinc}(x_0/a) \cdot \text{sinc}(x_1/a), & \text{if } \max(|x_0|, |x_1|) < a, \\ 0, & \text{if } \max(|x_0|, |x_1|) \geq a, \end{cases} \qquad (27)$$

We can now define the pseudo-rotation operator $\mathbf{R}_\alpha^*[Z]$ as a simple convolution with another filter that resembles $h_R$:

$$\mathbf{R}_\alpha^*[Z] = 1/s^2 \cdot \text{Ⅲ} \odot (h_R^* * Z) = H_R^* * Z,$$
$$h_R^* = \left(\psi * \mathbf{r}_\alpha[\psi]\right) \odot (w_L^+ * \mathbf{r}_\alpha[w_L^+]), \qquad (28)$$

where the discrete version $H_R^*$ is obtained from $h_R^*$ using Equation 6.

Finally, we define the valid region $\mathcal{V}$ the same way as in Appendix E.2: the set of pixels for which both filter footprints fall within the bounds of the corresponding original images.

# F    Implementation details

We implemented our alias-free generator on top of the official PyTorch implementation of StyleGAN2-ADA, available at `https://github.com/NVlabs/stylegan2-ada-pytorch`. We kept most of the details unchanged, including discriminator architecture [16], weight demodulation [16], equalized learning rate for all trainable parameters [13], minibatch standard deviation layer at the end of the discriminator [13], exponential moving average of generator weights [13], mixed-precision FP16/FP32 training [14], non-saturating logistic loss [8], $R_1$ regularization [18], lazy regularization [16], and Adam optimizer [17] with $\beta_1 = 0$, $\beta_2 = 0.99$, and $\epsilon = 10^{-8}$.

We ran all experiments on NVIDIA DGX-1 with 8 Tesla V100 GPUs using PyTorch 1.7.1, CUDA 11.0, and cuDNN 8.0.5. We computed FID between 50k generated images and all training images using the official pre-trained Inception network, available at `http://download.tensorflow.org/models/image/imagenet/inception-2015-12-05.tgz`

Our implementation and pre-trained models are available at `https://github.com/NVlabs/stylegan3`

## F.1    Generator architecture

**Normalization (configs D–R)**    We have observed that eliminating the output skip connections in StyleGAN2 [16] results in uncontrolled drift of signal magnitudes over the generator layers. This does not necessarily lead to lower-quality results, but it generally increases the amount of random variation between training runs and may occasionally lead to numerical issues with mixed-precision training. We eliminate the drift by tracking a long-term exponential moving average of the input signal magnitude on each layer and normalizing the feature maps accordingly. We update the moving

average once per training iteration, based on the mean of squares over the entire input tensor, and freeze its value after training. We initialize the moving average to 1 and decay it at a constant rate, resulting in 50% decay per 20k real images shown to the discriminator. With this explicit normalization in place, we have found it beneficial to slightly adjust the dynamic range of the output RGB colors. StyleGAN2 uses $-1$ and $+1$ to represent black and white, respectively; we change these values to $-4$ and $+4$ starting from config D and, for consistency with the original generator, divide the color channels by 4 afterwards.

**Transformed Fourier features (configs H–R)** We enable the orientation of the input features $z_0$ to vary on a per-image basis by introducing an additional affine layer (Figure 4b) and applying a geometric transformation based on its output. The affine layer produces a four-dimensional vector $\boldsymbol{t} = (r_c, r_s, t_x, t_y)$ based on $\mathbf{w}$. We initialize its weights so that $\boldsymbol{t} = (1, 0, 0, 0)$ at the beginning, but allow them to change freely over the course of training. To interpret $\boldsymbol{t}$ as a geometric transformation, we first normalize its value based on the first two components, i.e., $\boldsymbol{t}' = (r'_c, r'_s, t'_x, t'_y) = \boldsymbol{t}/\sqrt{r_c^2 + r_s^2}$. This makes the transformation independent of the magnitude of $\mathbf{w}$, similar to the weight modulation and demodulation [16] on the other layers. We then interpret the first two components as rotation around the center of the canvas $[0, 1]^2$, with the rotation angle $\alpha$ defined by $r'_c = \cos \alpha$ and $r'_s = \sin \alpha$. Finally, we interpret the remaining two components as translation by $(t'_x, t'_y)$ units, so that the translation is performed after the rotation. In practice, we implement the resulting geometric transformation by modifying the phases and two-dimensional frequencies of the Fourier features, which is equivalent to applying the same transformation to the continuous representation of $z_0$ analytically.

**Flexible layer specifications** In configs T and R, we define the per-layer filter parameters (Figure 4c) as follows. The cutoff frequency $f_c$ and the minimum acceptable stopband frequency $f_t$ obey geometric progression until the first critically sampled layer:

$$f_c[i] = f_{c,0} \cdot (f_{c,N}/f_{c,0})^{\min(i/(N-N_{\text{crit}}),1)}, \qquad f_t[i] = f_{t,0} \cdot (f_{t,N}/f_{t,0})^{\min(i/(N-N_{\text{crit}}),1)}, \quad (29)$$

where $N = 14$ is the total number of layers, $N_{\text{crit}} = 2$ is the number of critically sampled layers at the end, $f_{c,0} = 2$ corresponds to the frequency content of the input Fourier features, and $f_{c,N} = s_N/2$ is defined by the output resolution. $f_{t,0}$ and $f_{t,N}$ are free parameters; we use $f_{t,0} = 2^{2.1}$ and $f_{t,N} = f_{c,N} \cdot 2^{0.3}$ in most of our tests. Given the values of $f_c[i]$ and $f_t[i]$, the sampling rate $s[i]$ and transition band half-width $f_h[i]$ are then determined by

$$s[i] = \exp_2 \left\lceil \log_2 \left( \min(2 \cdot f_t[i], s_N) \right) \right\rceil, \qquad f_h[i] = \max(f_t[i], s[i]/2) - f_c[i]. \quad (30)$$

The sampling rate is rounded up to the nearest power of two that satisfies $s[i] \geq 2f_t[i]$, but it is not allowed to exceed the output resolution. The transition band half-width is selected to satisfy either $f_c[i] + f_h[i] = f_t[i]$ or $f_c[i] + f_h[i] = s[i]/2$, whichever yields a higher value.

We consider $f_c[i]$ to represent the output frequency content of layer $i$, for $i \in \{0, 1, \ldots, N - 1\}$, whereas the input is represented by $f_c[\max(i - 1, 0)]$. Thus, we construct the corresponding upsampling filter according to $f_c[\max(i - 1, 0)]$ and $f_h[\max(i - 1, 0)]$ and the downsampling filter according to $f_c[i]$ and $f_h[i]$. The nonlinearity is evaluated at a temporary sampling rate $s' = \max(s[i], s[\max(i - 1, 0)]) \cdot m$, where $m$ is the upsampling parameter discussed in Section 3.2 that we set to 2 in most of our tests.

### F.2 Hyperparameters and training configurations

We used 8 GPUs for all our training runs and continued the training until the discriminator had seen a total of 25M real images when training from scratch, or 5M images when using transfer learning. Figure 16 shows the hyperparameters used in each experiment. We performed the baseline runs (configs A–C) using the corresponding standard configurations: StyleGAN2 config F [16] for the high-resolution datasets in Figure 5, left, and ADA 256×256 baseline config [14] for the ablations in Figure 3 and Figure 5, right.

Many of our hyperparameters, including discriminator capacity and learning rate, batch size, and generator moving average decay, are inherited directly from the baseline configurations, and kept unchanged in all experiments. In configs C and D, we disable noise inputs [15], path length regularization [16], and mixing regularization [15]. In config D, we also decrease the mapping network depth to

| Parameter | Datasets (Figure 5, left) | | | Ablations at 256×256 | | |
|---|---|---|---|---|---|---|
| Config | B | T | R | A–C | D–T | R |
| Batch size | 32 | 32 | 32 | 64 | 64 | 64 |
| Moving average | 10k | 10k | 10k | 20k | 20k | 20k |
| Mapping net depth | 8 | 2 | 2 | 8 | 2 | 2 |
| Minibatch stddev | 4 | 4 | 4 | 8 | 4 | 4 |
| G layers | 15/17 | 14 | 14 | 13 | 14 | 14 |
| G capacity: $C_{\text{base}}$ | $2^{15}$ | $2^{15}$ | $2^{16}$ | $2^{14}$ | $2^{14}$ | $2^{15}$ |
| G capacity: $C_{\text{max}}$ | 512 | 512 | 1024 | 512 | 512 | 1024 |
| G learning rate | 0.0020 | 0.0025 | 0.0025 | 0.0025 | 0.0025 | 0.0025 |
| D learning rate | 0.0020 | 0.0020 | 0.0020 | 0.0025 | 0.0025 | 0.0025 |

| $R_1$ regularization $\gamma$ | | B | T | R |
|---|---|---|---|---|
| FFHQ-U | $256^2$ | 1.0 | 1.0 | 1.0 |
| FFHQ-U | $1024^2$ | 10.0 | 32.8 | 32.8 |
| FFHQ | $1024^2$ | 10.0 | 32.8 | 32.8 |
| METFACES-U | $1024^2$ | 10.0 | 16.4 | 6.6 |
| METFACES | $1024^2$ | 5.0 | 6.6 | 3.3 |
| AFHQV2 | $512^2$ | 5.0 | 8.2 | 16.4 |
| BEACHES | $512^2$ | 2.0 | 4.1 | 12.3 |

Figure 16: **Left:** Hyperparameters used in each experiment. **Right:** $R_1$ regularization weights.

2 and set the minibatch standard deviation group size to 4 as recommended in the StyleGAN2-ADA documentation. The introduction of explicit normalization in config D allows us to use the same generator learning rate, 0.0025, for all output resolutions. In Figure 5, right, we show results for path length regularization with weight 0.5 and mixing regularization with probability 0.5.

**Augmentation** Since our datasets are horizontally symmetric in nature, we enable dataset $x$-flip augmentation in all our experiments. To prevent the discriminator from overfitting, we enable adaptive discriminator augmentation (ADA) [14] with default settings for METFACES, METFACES-U, AFHQV2, and BEACHES, but disable it for FFHQ and FFHQ-U. Furthermore, we train METFACES and METFACES-U using transfer learning from the corresponding FFHQ or FFHQ-U snapshot with the lowest FID, similar to Karras et al. [14], but start the training from scratch in all other experiments.

**Generator capacity** StyleGAN2 defines the number of feature maps on a given layer to be inversely proportional to its resolution, i.e., $C[i] = C(s[i]) = \min(\text{round}(C_{\text{base}}/s[i]), C_{\text{max}})$, where $s[i]$ is the output resolution of layer $i$. Parameters $C_{\text{base}}$ and $C_{\text{max}}$ control the overall capacity of the generator; our baseline configurations use $C_{\text{max}} = 512$ and $C_{\text{base}} = 2^{14}$ or $2^{15}$ depending on the output resolution. Since StyleGAN2 can be considered to employ critical sampling on all layers, i.e., $f_c[i] = s[i]/2$, we can equally well define the number of feature maps as $C[i] = C(2f_c[i])$. These two definitions are equivalent for configs A–F, but in configs G–R we explicitly set $f_c[i] \le s[i]/2$, which necessitates using the latter definition. In config R, we double the value of both $C_{\text{base}}$ and $C_{\text{max}}$ to compensate for the reduced capacity of the $1\times1$ convolutions. In Figure 5, right, we sweep the capacity by multiplying both parameters by 0.5, 1.0, and 2.0.

**$R_1$ regularization** The optimal choice for the $R_1$ regularization weight $\gamma$ is highly dependent on the dataset, necessitating a grid search [16, 14]. For the baseline config B, we tested $\gamma \in \{1, 2, 5, 10, 20\}$ and selected the value that gave the best FID for each dataset. For our configs T and R, we followed the recommendation of Karras et al. [14] to define $\gamma = \gamma_0 \cdot N/M$, where $N = s_N^2$ is the number of output pixels and $M$ is the batch size, and performed a grid search over $\gamma_0 \in \{0.0002, 0.0005, 0.0010, 0.0020, 0.0050\}$. For the low-resolution ablations, we chose to use a fixed value $\gamma = 1$ for simplicity. The resulting values of $\gamma$ are shown in Figure 16, right.

**Training of config R** In this configuration, we blur all images the discriminator sees in the beginning of the training. This Gaussian blur is executed just before the ADA augmentation. We start with $\sigma = 10$ pixels, which we ramp to zero over the first 200k images. This prevents the discriminator from focusing too heavily on high frequencies early on. It seems that in this configuration the generator sometimes learns to produce high frequencies with a small delay, allowing the discriminator to trivially tell training data from the generated images without providing useful feedback to the generator. As such, config R is prone to random training failures in the beginning of the training without this trick. The other configurations do not have this issue.

### F.3 G-CNN comparison

In Figure 5, bottom, we compare our config R with config T extended with $p4$-symmetric group convolutions [4, 5]. $p4$ symmetry makes the generator equivariant to 0°, 90°, 180°, and 270° rotations, but not to arbitrary rotation angles. In practice, we implement the group convolutions by extending all intermediate activation tensors in the synthesis network with an additional group dimension of size 4 and introducing appropriate redundancy in the convolution weights. We keep the input layer

| Item | Number of training runs | GPU years (Volta) | Electricity (MWh) |
|---|---|---|---|
| Early exploration | 233 | 18.02 | 42.45 |
| Project exploration | 1207 | 48.93 | 118.13 |
| Setting up ablations | 297 | 13.30 | 32.48 |
| Per-dataset tuning | 63 | 4.54 | 13.28 |
| Producing results in the paper | 53 | 5.26 | 14.35 |
|    StyleGAN3-R at 1024×1024 | 1 | 0.30 | 0.87 |
|    Other runs in the dataset table | 17 | 2.35 | 6.88 |
|    Ablation tables | 35 | 2.61 | 6.60 |
| Results intentionally left out | 23 | 1.72 | 3.93 |
| Total | 1876 | 91.77 | 224.62 |

Figure 17: Computational effort expenditure and electricity consumption data for this project. The unit for computation is GPU-years on a single NVIDIA V100 GPU — it would have taken approximately 92 years to execute this project using a single GPU. See the text for additional details about the computation and energy consumption estimates. **Early exploration** includes early training runs that affected our decision to start this project. **Project exploration** includes training runs that were done specifically for this project, leading to the final StyleGAN3-T and StyleGAN3-R configurations. These runs were not intended to be used in the paper as-is. **Setting up ablations** includes hyperparameter tuning for the intermediate configurations and ablation experiments in Figures 3 and 5. **Per-dataset tuning** includes hyperparameter tuning for individual datasets, mainly the grid search for $R_1$ regularization weight. **Config R at 1024×1024** corresponds to one training run in Figure 5, left, and **Other runs in the dataset table** includes the remaining runs. **Ablation tables** includes the low-resolution ablations in Figures 3 and Figure 5. **Results intentionally left out** includes additional results that were initially planned, but then left out to improve focus and clarity.

unchanged and introduce the group dimension by replicating each element of $z_0$ four times. Similarly, we eliminate the group dimension after the last layer by computing an average of the four elements. $p4$-symmetric group convolutions have $4\times$ as many trainable parameters as the corresponding regular convolutions. To enable an apples-to-apples comparison, we compensate for this increase by halving the values of $C_{\text{base}}$ and $C_{\text{max}}$, which brings the number of parameters back to the original level.

## G    Energy consumption

Computation is an essential resource in machine learning projects: its availability and cost, as well as the associated energy consumption, are key factors in both choosing research directions and practical adoption. We provide a detailed breakdown for our entire project in Table 17 in terms of both GPU time and electricity consumption. We report expended computational effort as single-GPU years (Volta class GPU). We used a varying number of NVIDIA DGX-1s for different stages of the project, and converted each run to single-GPU equivalents by simply scaling by the number of GPUs used.

We followed the Green500 power measurements guidelines [7]. The entire project consumed approximately 225 megawatt hours (MWh) of electricity. Approximately 70% of it was used for exploratory runs, where we gradually built the new configurations; first in an unstructured manner and then specifically ironing out the new StyleGAN3-T and StyleGAN3-R configurations. Setting up the intermediate configurations between StyleGAN2 and our generators, as well as, the key parameter ablations was also quite expensive at ~15%. Training a single instance of StyleGAN3-R at 1024×1024 is only slightly more expensive (0.9MWh) than training StyleGAN2 (0.7MWh) [16].