# OpenReview forum: "Alias-Free Generative Adversarial Networks"
_NeurIPS.cc/2021/Conference — NeurIPS 2021 Oral_

### Official Review · Reviewer_6RJP · 2021-07-11

**Rating:** 8
**Confidence:** 5

**Summary:**

This paper analyzed the aliasing and equivariance issues of standard GANs components through the lens of signal processing and proposed several corresponding techniques to modify the StyleGAN2 generator to make it better suited to video and animation tasks. The visual results are impressive and will benefit a variety of following works.

**Limitations And Societal Impact:**

The authors adequately addressed the limitations and potential negative societal impact of their work.

**Main Review:**

Overall, this is a high-quality paper with solid contributions to be published at NeurIPS.

**Originality**: This paper makes the first step in solving aliasing and equivariance issues in GANs using a series of newly proposed methods. Both the tasks and methods are new to the GAN community.

**Quality**: This paper is of high quality. Both the theoretical reasoning and the experimental results are convincing. However, it could be better to include experiments or some discussions on the following points:
1. The method is built on the assumption of limited bandwidth signals, whose valid bandwidth is restricted by the sampling rate or image resolution. So how does the proposed method work on low-resolution images where the sampling rate is more challenging?
2. What exactly happens when the training dataset contains very high-frequency signals? Will it simply fail (as indicated by the authors) or will the generated images look like the ones that have been passed through a low-pass filter? This may help better understand the limitation of the proposed method.

**Clarity**: This paper is well-organized but some parts of it need more explanation:
1. There seems to be a gap in motivation. The “texture sticking” phenomenon, positional references and equivariance are easy to understand, but how are they related to aliasing? I do not see strong evidence showing that “texture sticking” is caused by aliasing. So why is the paper named “Aliasing-free GANs” but not something like “Equivariant GANs”? Equivariance seems to be a more straightforward motivation.
2. How do the modulation and demodulation influence the equivariance?
3. The visualization in Figure 6 looks impressive. However, the explanation of how it is generated and how to interpret it is not enough (e.g. why the proposed method yield the grid/wave-like patterns?).

**Significance**: The results are impressive and the contributions are solid. I believe it will attract many follow-up works in GAN-based video and animation synthesis and processing.




**Time Spent Reviewing:**

10

---

> ### Author Response · Authors · 2021-08-10
> **Reply to Reviewer 6RJP**
>
> Thank you for the review. Answers to specific questions:
>
> Q: What exactly happens when the training dataset contains very high-frequency signals? Will it simply fail (as indicated by the authors) or will the generated images look like the ones that have been passed through a low-pass filter? This may help better understand the limitation of the proposed method.
> Q: The method is built on the assumption of limited bandwidth signals, whose valid bandwidth is restricted by the sampling rate or image resolution. So how does the proposed method work on low-resolution images where the sampling rate is more challenging?
>
> The results do not look blurry. We have compared the frequency content between our generator and the training data, and high frequencies are properly reproduced. We can add this comparison in the supplement. Of the results in the paper, AFHQ is full of very high frequencies, and they get properly replicated.
>
> The passage around lines L340--343 is carelessly written, and we apologize for that. Let us consider an example. Assume that we have high-resolution black-and-white cartoons as training data, and we pre-process these (incorrectly) using point sampling, leading to training images where almost all pixels are either black or white and the edges are jagged. This kind of badly aliased training data is difficult for GANs in general, but it is especially at odds with equivariance: on one hand, we should be able to translate the output smoothly by subpixel amounts, but on the other hand, edges should remain jagged and pixels only black/white, to remain faithful to the training data. This same issue can also arise with retro pixel graphics, where the jagged stair-step edges are a defining feature of the aesthetic, and it may thus be necessary for the synthesis to be aware of the pixel grid. We are happy to clarify this in a revision.
>
> We have not experimented with tiny images like CIFAR, but to the best of our knowledge there should be no problems.
>
>
> Q1. There seems to be a gap in motivation. The “texture sticking” phenomenon, positional references and equivariance are easy to understand, but how are they related to aliasing? I do not see strong evidence showing that “texture sticking” is caused by aliasing. So why is the paper named “Aliasing-free GANs” but not something like “Equivariant GANs”? Equivariance seems to be a more straightforward motivation.
>
> Our goal is to nudge GANs to do proper hierarchical synthesis. In the introduction we identify three reasons that can prevent this from happening: borders, noise inputs, and aliasing. Looking at video 5, texture sticking remains clearly visible when the two first sources have been eliminated, and fixing aliasing then solves the problem.
>
> As for the paper’s title, we specifically focus on aliasing in this work, and thus felt that a more narrow and specific title was appropriate. Also, there are earlier papers that refer to the wider context of equivariance in their title but approach it from quite different directions.
>
>
> Q: How do the modulation and demodulation influence the equivariance?
>
> They don’t, as they modify the convolution weights globally.
>
>
> Q: The visualization in Figure 6 looks impressive. However, the explanation of how it is generated and how to interpret it is not enough (e.g. why the proposed method yield the grid/wave-like patterns?).
>
> Once the alternative sources of positional information are removed, the positional information must become encoded in the feature maps in one form or another. The contribution of Figure 6 is to show that coordinate system -like feature maps really do emerge. A deeper analysis of these feature maps and the way they are utilized by the network will be an interesting topic of future research.

---

### Official Review · Reviewer_1gcw · 2021-07-13

**Rating:** 9
**Confidence:** 4

**Summary:**

This paper explores in detail the intrinsic properties of StyleGAN generators that lead to a variety of aliasing artifacts. The source of these artifacts is traced to several distinct phenomena, which are pointwise addressed via a lens of analysis in the continuous signal space, relying heavily on fundamental signal processing techniques. The resulting solutions are applied to a StyleGAN generator, resulting in an “alias-free” model that is better able to handle effects like pose variation while still maintaining the sample quality of the baseline.

**Limitations And Societal Impact:**

Yes. It's important to note that StyleGAN models in particular have been used to fuel bot armies and disinformation campaigns through the use of generated faces, and the authors acknowledge this.

**Main Review:**

This is an excellent paper based around a solid premise, with strong empirical results against relevant and up-to-date baselines. The analysis is easy to follow and the proposed methods are sensible. This paper is a clear accept without need for revision; I feel that (barring actual mistakes, which I am not able to spot after reading through in detail) seeking out minor points to trip the authors up on would be a waste of time for all involved.

**Time Spent Reviewing:**

3

---

> ### Author Response · Authors · 2021-08-10
> **Reply to Reviewer 1gcw**
>
> Thank you for the review.

---

### Official Review · Reviewer_gQig · 2021-07-15

**Rating:** 10
**Confidence:** 5

**Summary:**

Starting from the observation that current generative models in fact do not synthesize images in a natural hierarchical manner (which they call “texture sticking”), this paper defines the problem systematically, analyze the root of it (unintentional information leaking and aliasing), and proposes a set of simple architectural solutions that are motivated from the signal processing tools.
The key component of the approach are:
- pointing out the presence of “texture sticking” phenomenon that much of the fine details of synthesized images appear to be fixed in the absolute pixel coordinates, not the relative position of the underlying objects being drawn.
- analyzing signals that flow through a network in a continuous domain and interpreting the actual, discrete features as merely a convenient sampled encoding of it
- fixing the operations (up/down sampling and nonlinearity) that are the root of the aliasing by applying a proper low-pass filtering.
After applying these fixes, the generator becomes translation- and/or rotation- invariant and provides authentic 2D images as if there is actual 3D model and it is capturing pictures in different views.


**Ethical Concerns:**

No comment.

**Limitations And Societal Impact:**

No comment.

**Main Review:**

This is a very nice work with impressive results, a great progress achievement in the field of image generation. Very well written.
In particular, I am very impressed with how deeply the authors investigate the deficiencies of the current state-of-the-art, which could have been easily overlooked. The way that the authors analyze the problem is very convincing, and not only do the theoretical analyses provide deep insights but the empirical results shows big advances by fixing the issues at a fundamental level. I believe these insights will be very useful for the future developments of the field.
Except a few minor things for clarification, I recommend the paper to be accepted as is.

1) If the ReLU-type of nonlinearity is the problem, would using other smooth type of nonliearity (like swish) function give better result?
2) In config R, every 3x3 convolution is now replaced to 1x1. Then, is it correct to understand that the network is now entirely relying on the interpolation function (upsamlpling part) for learning the relationship between the neighboring pixels?

2) Line 135, “convolving it with a discretized downsampling filter …” I think this should be changed to low-pass filter.
3) Line 34, “faint after-images of the pixel grid …”: what does this mean? It was not clear for me even after reading the entire paper. Could you please clarify this a bit more in the paper?
4) Line 93, “In the remainder of this paper we shall interpret …” In the remainder of this paper, we shall interpret … (comma is missing)
5) Line 153, “This discrete operation cannot be realized without temporarily entering the continuous
representation.” This was not obvious to me. Isn’t the downsampling operation has the same issue (since it needs to be convolved with low-pass filter in continuous domain)? Can you elaborate a bit more?
6) Line 158 to Line 161, Does this mean that if we limit the range of novel frequencies very strictly, the generator won't be able to add any fine details?

Overall, as stated above, the paper is well written. However, considering the impact of the paper to the field, it would nice if the authors elaborate the details (that I pointed here) a bit more for the ones who are not familiar with signal processing context.

---

update after the rebuttal: I am satisfied with the authors' response and I keep my score.

Some questions I actually forgot to ask:
1) "Recall that sampling rate s equals the size of the canvas in pixels, given our definitions in..." Here, what is the (physical) size of the canvas then? How do you set it?
2) "but as a separable filter it compromises rotation equivariance in particular.": why?
3) "they contain strong overlays.": what does this "overlays" mean?

---

**Time Spent Reviewing:**

8

---

> ### Author Response · Authors · 2021-08-10
> **Reply to Reviewer gQig**
>
> Thank you for the review. Answers to specific questions:
>
> Q: If the ReLU-type of nonlinearity is the problem, would using other smooth type of nonliearity (like swish) function give better result?
>
> The problem is not limited to ReLU, as any pointwise nonlinear function will widen the range of frequencies in the signal, and thus require bandlimiting to suppress aliasing. Intuitively, smoother nonlinearities such as swish should yield lower amplitudes in the aliasing frequency band. This might make it sufficient to have a bit less attenuation in the bandlimiting filters, but not remove the need altogether.
>
>
> Q: In config R, every 3x3 convolution is now replaced to 1x1. Then, is it correct to understand that the network is now entirely relying on the interpolation function (upsamlpling part) for learning the relationship between the neighboring pixels?
>
> Correct, in this configuration only the up and down sampling operations spread information between pixels.
>
>
> Q: Line 34, “faint after-images of the pixel grid …”: what does this mean? It was not clear for me even after reading the entire paper. Could you please clarify this a bit more in the paper?
>
> Consider nearest neighbor upsampling. If we have a 4x4 image and upsample that to 8x8, the original pixels will be very clearly visible, allowing one to reliably distinguish between even and odd pixels. Since the same is true on all scales, this (leaked) information makes it possible to reconstruct even the absolute pixel coordinates. When we switch to a better filter such as bilinear, the clues get less pronounced, but are nevertheless evident for the generator. The observation that very high-quality filters with extreme suppression of aliasing frequencies are needed before the generator lets go of these clues is one of the main surprises in our work.
>
>
> Q: Line 153, “This discrete operation cannot be realized without temporarily entering the continuous representation.” This was not obvious to me. Isn’t the downsampling operation has the same issue (since it needs to be convolved with low-pass filter in continuous domain)? Can you elaborate a bit more?
>
> It is true that, in principle, the downsampling result needs to be convolved with a low-pass filter in the continuous domain. However, because we know that the frequency content is bounded -- ideal downsampling only cuts away part of the frequency spectrum -- this can be implemented exactly in the discrete domain. The same is not true for nonlinearity because it can smear the spectrum in unpredictable ways and expand it to the aliasing frequency band.
>
>
> Q: Line 158 to Line 161, Does this mean that if we limit the range of novel frequencies very strictly, the generator won't be able to add any fine details?
>
> This is correct -- hence the increasing cutoff frequency throughout the generator.
>
>
> Thanks for pointing out the minor issues on lines 93 and 135, we will fix those as well.

---

### Official Review · Reviewer_B84m · 2021-07-20

**Rating:** 6
**Confidence:** 4

**Summary:**

The paper notes that the current GAN architectures do not synthesize images in a natural hierarchical manner. Finer details in the images are dependent on pixel coordinates, which is undesirable. This is evident from the internal representations learned by the network. The authors attribute this problem to aliasing and propose changes to get rid of it. Specifically, they propose changes to upsampling, downsampling, and non-linearity to make the features rotation and translation invariant. Finally, the authors apply those changes to StyleGAN2 architecture and show comparable FID scores and improved translation and rotation invariance.

**Limitations And Societal Impact:**

Even if the proposed changes apply to any convolutional architecture, the tests have been done only on StyleGAN2, which limits the empirical soundness of the results. The analysis is limited to GANs without attention layers. Either this should be specified in the limitations or experiments related to attention layers should be added.

**Main Review:**

Overall, the paper is well motivated and does a good job of explaining the problem. Authors introduce "non-critical sampling" and the "flexible layers" which clearly show to increase desirable invariances for GANs and increase FID scores. The paper is moderately well written, clarifications.

Here are some questions that need discussion:
Why would the handling of boundaries and upsampling hurt FID? Why would filtered non-linearities hurt FID?  Does the change from D to E keep the filter size to be the same? Wondering if the increase in translation invariance with E is causes just by using a larger filter size! Why does adding noise is contrary to the goal of this paper? Can they be treated as inputs, and hence, they should be translated (or rotated) to compute translation (or rotation) invariance metrics?

The biggest contributor to the performance seems to come from “non-critical sampling” and the “flexible layers”. It will be valuable for readers to know how does the performance look with just these two changes.



**Time Spent Reviewing:**

6

---

> ### Author Response · Authors · 2021-08-10
> **Reply to Reviewer B84m**
>
> Thank you for the review. Answers to specific questions:
>
> Q: Why would the handling of boundaries and upsampling hurt FID? Why would filtered non-linearities hurt FID?
>
> We suspect that the explanation relates to the large-scale “mode flip” between StyleGAN2 and Alias-Free GAN. In this ablation, the early configs rely heavily on the unintentional positional information, whereas in the later configs the generator ends up inventing its own coordinate systems (Fig.6). These “phase signals” do not yet appear in the middle configs E & F, but we have already started to constrain the frequencies that the feature maps can contain, limiting the possibilities to carry information through aliasing. Thus, the generator may have less room for maneuvering.
>
> Subsequent configs adjust the distribution of feature maps in the network (without increasing the overall capacity) to better accommodate the needs of the new design. In that sense, one could argue that a full hyperparameter search per intermediate config might yield better results. However, we feel that it is important to show in the ablation that there are necessary intermediate steps along the way that in isolation make the results worse, and that many simultaneous changes are needed to make the new design work well.
>
>
> Q: Does the change from D to E keep the filter size to be the same? Wondering if the increase in translation invariance with E is causes just by using a larger filter size!
>
> The filter size does increase in this step. In practice, it makes sense to bundle the increased footprint, transition from bilinear to Kaiser, and improved border treatment into a single step. If we simply made the bilinear filter wider, it would attenuate high frequencies too aggressively and thus hamper the generator’s ability to synthesize fine details. If we switched to the wide Kaiser filter without fixing the borders, we would see no improvement in EQ-T as the benefit from reduced aliasing would effectively get nullified by the stronger ringing artifacts around the borders.
>
> Fig.3 (right) includes an additional sweep for the filter size and Fig.5 (right) tests a few alternative filters.
>
>
> Q: Why does adding noise is contrary to the goal of this paper? Can they be treated as inputs, and hence, they should be translated (or rotated) to compute translation (or rotation) invariance metrics?
>
> If the goal was simply to, e.g., translate a generated image, one could indeed obtain the correct result by translating the noise inputs. But our goal is considerably broader than that: we want to force the generator to perform proper hierarchical synthesis, where finer details implicitly follow the movement of the coarser ones -- even when the exact transformation is not known in advance. For example, if you wanted to turn a generated head by modifying the latent, you wouldn’t know how the noise inputs should be transformed.
>
>
> Q: The biggest contributor to the performance seems to come from “non-critical sampling” and the “flexible layers”. It will be valuable for readers to know how does the performance look with just these two changes.
>
> Please note that Figure 3 is not a collection of orthogonal changes; it’s a progression where each configuration builds on the previous one. For example, config T adjusts the Kaiser filter’s parameters, but we wanted to introduce that filter in an earlier step to provide additional insights. The right-hand sides of Figures 3 and 5 study the importance of various choices in the context of our final models (configs T and R).
>
>
> Q: Even if the proposed changes apply to any convolutional architecture, the tests have been done only on StyleGAN2, which limits the empirical soundness of the results. The analysis is limited to GANs without attention layers. Either this should be specified in the limitations or experiments related to attention layers should be added.
>
> This is a good point, and we will gladly clarify it in a revision. So far, attention has been incorporated in GANs in two primary ways. Early works, such as SAGAN, simply introduce one self-attention layer in the middle of a traditional generator, and this could likely be dealt with similarly to non-linearities by temporarily switching to higher resolution -- although the time complexity of the attention layer may make this somewhat challenging in practice. More recent techniques that start with a tokenizing transformer (e.g., VQGAN) may actually be at odds with equivariance goals. Whether it’s possible to make them equivariant is an important open question for the community.

---

### Decision · Program_Chairs · 2021-09-27

**Decision:**

Accept (Oral)

**Comment:**

The paper identifies the "texture sticking" problem in image synthesis GANs, argues that this is caused by aliasing in the convolutions, proposes architectural modifications to fix the aliasing, and demonstrates strong improvements in equivariance / smooth transitions. All reviewers agreed that the results were impressive. Separately and equally importantly, all reviewers agreed that the problem was clearly motivated and multiple reviewers (gQig, 1gcw) also agreed that the analysis was insightful. Papers which succeed on both of these fronts are rare; I recommend acceptance.